# Recent Progress on Relevant microRNAs in Autism Spectrum Disorders

**DOI:** 10.3390/ijms21165904

**Published:** 2020-08-17

**Authors:** Xingwang Wu, Wanran Li, Yun Zheng

**Affiliations:** Yunnan Key Lab of Primate Biomedicine Research, Institute of Primate Translational Medicine, Kunming University of Science and Technology, Kunming 650500, China; wxwwxw7311696@163.com (X.W.); wanyou0764@163.com (W.L.)

**Keywords:** autism spectrum disorder (ASD), miRNA, miRNA target, biomarkers, therapy

## Abstract

Autism spectrum disorder (ASD) is a neurodevelopmental disorder whose pathogenesis is unclear and is affected by both genetic and environmental factors. The microRNAs (miRNAs) are a kind of single-stranded non-coding RNA with 20-22 nucleotides, which normally inhibit their target mRNAs at a post-transcriptional level. miRNAs are involved in almost all biological processes and are closely related to ASD and many other diseases. In this review, we summarize relevant miRNAs in ASD, and analyze dysregulated miRNAs in brain tissues and body fluids of ASD patients, which may contribute to the pathogenesis and diagnosis of ASD.

## 1. Introduction

Autism spectrum disorder (ASD) is a developmental nervous system disease with high incidence. ASD occurs in early childhood, so it is named as children’s autism [1]. ASD mainly begins before the age of three, and the most obvious stage of ASD behavior is 2–5 years old [2]. It has been reported that the ASD incidence of men is higher than that of women [3]. The incidence of ASD worldwide is about 1%, which is relatively common [4,5], and it was estimated that 1 in 59 children was affected by ASD in 2018 [6]. The main manifestations of ASD are interpersonal barriers, language communication barriers, and behavioral stereotypes [7,8]. Although the cause of ASD is still unclear, it is currently believed that ASD is related to genetic and environmental factors [9,10,11], as well as neurotransmitter systems, such as 5-hydroxytryptamine, dopamine, and glutamate [12]. Among these, genetics occupies the main factor, accounting for about 80% of the risk of disease [13,14,15,16]. Some studies reported that genetic variations of genes may be related to the pathogenesis of ASD and affect synaptic plasticity [4,17,18]. Other studies demonstrated that ASD is associated with impaired synaptic transmission and connection at the nervous system level [19,20]. It has also been reported that epigenetic modification can regulate the underlying pathogenesis of ASD at the gene and environmental levels [21]. However, the specific pathogenic mechanism of ASD is still unclear.

Psychological education, nursing coordination, and improved cognitive behavior therapy [22] are the current treatment options for ASD. Since the treatment efficiency is low, miRNAs were selected as potential biomarkers to improve the diagnosis, prognosis, and treatment of ASD [23,24,25]. Furthermore, many studies have reported that children and adults with ASD are susceptible to autoimmune diseases such as asthma or atopic dermatitis [26,27,28,29,30]. Compared with 50 years ago, the life quality of ASD patients has been significantly improved through the effort that has been made by the whole of society. More and more adult patients with ASD can live independently. However, there are still some ASD patients who have difficulty with independent living. Therefore, we need more help and care from the whole of society to improve the living situation of ASD patients [31]. The total number of autistic patients in the world has reached 67 million [32], which is larger than the sum of AIDS, cancer, and diabetes patients. With the increase in morbidity, ASD brings a huge social and economic burden to families of ASD patients and even the whole of society. Therefore, it is urgent to carry out basic research for ASD.

The miRNAs are a kind of single-stranded non-coding small RNA with 20-22 nucleotides, encoded by the genomes of animals and plants, and play important roles in regulating gene expression at the post-transcriptional level [33]. The processing of miRNA is mainly composed of two steps. The first step is to generate miRNA precursors (pre-miRNAs) about 70 nucleotides from long endogenous primary transcripts of miRNAs (pri-miRNAs) in the nucleus by Drosha and DGCR8 [34,35]. Exportin-5 then exports pre-miRNAs to the cytoplasm [35,36]. In the second step, the pre-miRNAs are processed into mature miRNAs by the Dicer enzyme in the cytoplasm [37]. The miRNAs guide the RNA-induced silencing complex (RISC), normally with an Argonaute (AGO) protein, to the binding site on their target mRNAs through sequence matching, thereby causing degradation, inhibiting translation, or inducing cleavage of their target transcripts [38,39]. 

The miRNAs are involved in almost all important biological processes [40,41,42,43,44,45] and play key roles in various human diseases [46,47,48,49]. They are important regulators of brain development and neuronal function, and are associated with a variety of nervous system diseases [50,51]. At present, most research of miRNAs in ASD have mainly focused on the dysregulated miRNAs in the brain tissues and body fluids of ASD patients and only a few studies have paid attention to the functions of miRNAs in ASD. We summarize and discuss recent studies on miRNAs related to ASD, and explore potential methods for diagnosing and treating ASD based on miRNAs. For information on miRNAs in ASD animal models, please read Schepici et al., 2019 [52]; for synergy of miRNAs in ASD and allergic dermatitis, refer to Tonacci et al., 2019 [53]; for mutations of miRNAs in ASD patients, please refer to Fregeac et al., 2016 [54]; for dysregulation of circRNA in patients with ASD and the corresponding pathogenic role of circRNA–microRNA–mRNA regulation axes in ASD, please refer to Chen et al., 2020 [55]; for information on dysregulated miRNAs in ASD patients as potential diagnostic biomarkers and potential therapeutic targets, please refer to Anitha and Thanseem, 2015 [56].

## 2. The miRNAs with Characterized Functions in ASD

The mitogen-activated protein kinases (MAPK) signaling pathway, in which the candidate target genes of both let-7a and let-7d are involved, is directly or indirectly associated with the physiopathology of ASD [57]. Ghahramani Seno et al. previously reported that MAPK may be involved in the development of ASD [58]. Moreover, a previous study revealed that embryonic development may be also relevant to ASD [59]. It has been found that let-7a is highly expressed in early human embryonic tissues and suddenly decreases thereafter, which suggests the key role let-7a plays in early embryonic development [60]. Therefore, through the embryonic development process, let-7a may be indirectly associated with ASD [57].

As shown in Table 1, miR-21-3p is overexpressed in cortex regions of ASD patients, and miR-21-3p inhibits multiple genes in the M16 gene module on 15q11-13, a genomic region rich in neuronal and synaptic genes. These M16 module genes include DLGAP1 (a scaffold protein that interacts with SHANK3), as well as PAFAH1B1/LIS1 and DYNC1I1 [24]. These three genes play key roles in the migration of neurons [61]. In conclusion, the M16 module gene is related to the neuronal migration and synapses of ASD [24]. Besides, the overexpression of miR-21-3p leads to a significant decrease in the expression of PCDH19, which is related to cognitive impairment, and the mutation of PCDH19 will affect ASD [62]. Therefore, miR-21-3p plays a role in regulating neuronal and synaptic functions in patients with ASD by regulating the M16 gene module [24]. Oxytocin receptor (OXTR) is related to many social behaviors [63], and the social behaviors of ASD can be significantly improved by oxytocin therapy [64,65,66]. Furthermore, ASD is significantly related to single-nucleotide polymorphisms (SNPs) of OXTR [67]. It is interesting that another miRNA in pre-miR-21, miR-21-5p, is also relevant in ASD. In the ASD brain, overexpressed miR-21-5p targets the OXTR gene, and inhibits the translation of OXTR [68]. In summary, miR-21-5p may aggravate the phenotype of ASD by reducing the expression of OXTR in the ASD brain [68].

As shown in Table 1, muscular dystrophy and hypotonic and muscular weakness are associated with ASD [77,78,79]. Human brain-specific miR-29b represses the expression of COL6A2. The mutation of COL6A2 results in a reduction of COL6A2 transcripts, thereby destroying the stability of collagen and leading to a decrease in muscle strength [80]. It is speculated that the inhibition of COL6A2 caused by the up-regulation of miR-29b may be one of the potential genetic mechanisms of ASD muscle diseases and dyskinesia [59]. ID3 is the target of miR-29b [59]. In addition, miR-29b is associated with circadian rhythm signals, and studies have reported that ASD is associated with circadian rhythm disturbances [59]. The overexpression of miR-29b leads to the down-regulation of the ID3 transcript [59], which is also a neuronal target of MeCP2, and MeCP2 is the causative gene of Rett syndrome [81].

As shown in Table 1, it has been revealed that candidate target genes of miR-103a-3p are associated with dysfunctional physiological pathways of ASD, including various pathways that affect central nervous system development [57]. A candidate target gene of miR-103a-3p is brain-derived neurotrophic factor (BDNF), which directly or indirectly participates in ASD and plays a key role in neuronal differentiation and synapses [69,82]. In addition, BDNF was increased in some ASD patients and animal models [83,84]. The miRNAs miR-103 and miR-107 are highly expressed in brain tissue [85], and the specific targets of miR-103 and miR-107 in the brain are unclear, but miR-103 and miR-107 are related to lipid metabolism [85]. In addition, ASD is related to fatty acid metabolism [59], and studies have reported that abnormal lipid and fatty acid metabolism in ASD may be related to the imbalance of miR-103 and miR-107 [59].

It has been reported that miR-146a is up-regulated in various neural developmental disorders [70], and is highly expressed in brain regions such as the cortex, hippocampus, and amygdala, which are key structures of high cognitive ability [86]. The miRNA miR-146a inhibits the expression of neuronal-specific genes, Nlgn1, and Syt1, which are related to glial cell differentiation [87]. In the brain of the mouse model, the overexpression of miR-146a induces damage in neuronal dendrites, resulting in shrinking dendrites, thereby increasing the glutamate uptake capacity of astrocytes [70]. Finally, the imbalance of glutamate homeostasis impairs synaptic transmission in ASD [70]. 

As shown in Table 1, GRIA3, which encodes the core subunit of the α-amino-3-hydroxy-5-methyl-4-isoxazolepropionic acid (AMPA) receptor (AMPAR), an important factor in endocytosis, is a target gene of miR-146a [88]. MAP1B regulates AMPAR and is also targeted by miR-146a [89]. Therefore, the up-regulation of miR-146a in ASD neurons inhibits the translation of MAP1B and reduces the expression of GRIA3, which impairs the endocytosis of AMPA and disrupts synaptic transmission [70]. 

As shown in Table 1, the up-regulation of miR-146a in ASD also inhibits the expression of KCNK2, which plays a key role in the excitation of cortical neurons [90]. In addition, the knockdown of KCNK2 impairs the neuronal migration of mouse cerebral cortex [91], which is consistent with the decrease in KCNK2 gene expression in ASD that inhibits neuronal migration [70]. 

The up-regulation of miR-146a occurs in the early childhood brain of ASD [92]. In H9 human neural stem cells (H9 hNSCs), the overexpression of miR-146a results in neurite outgrowth, branching enhancement, the imbalance between neural progenitor cell renewal, and neuronal differentiation, which may ultimately affect ASD brain dysfunction [92]. Autistic patients have signs of nerve inflammation, immune abnormality, and changes in the inflammatory response throughout the whole of their lives [93]. Furthermore, the expression of miR-146a contributes to neuroinflammation in the brain of ASD patients, suggesting its role in immune system regulation [53].

The miRNA miR-146a is one of the most commonly dysregulated miRNAs in developmental brain disorder (DBD) patients, including ASD [54]. The down-regulation of miR-146a in the hippocampus of adult mice leads to damage to neural progenitor cell differentiation, and eventually cause serious learning and memory deficits [94]. These results demonstrated that miR-146a is also related to brain functions in adults. In miR-146^-/-^ mice, several phenotypes of DBD patients, such as impaired neurogenesis, abnormal brain anatomy, and memory deficits, were noticed, suggesting that the dysregulation of miR-146a contributes to the pathogenesis of DBD patients [94]. 

The miRNA miR-146a may play a key role in macrophage polarization [95]. The activation of M1 macrophages is inhibited by miR-146a, and miR-146a inhibits the Notch1 gene, which results in the enhanced activation ability of M2 macrophages. In addition, miR-146a activates PPARγ to promote the activation of M2 macrophages [95]. M1 macrophages and M2 macrophages are beneficial to the pathogenesis of inflammation and degenerative diseases [96,97]. In summary, miR-146a may be a potential therapeutic target for inflammatory diseases in the future [95].

NUMB plays a key role in the asymmetric division of neural progenitor cells and the determination of cell fate [98,99]. The miRNA miR-146a targets the NUMB gene, thereby activating Sonic hedgehog (SHH) signaling expression [100]. In addition, the maintenance and development of homeostasis in the intestine are related to SHH signaling [101,102], and miR-146a or SHH signaling has a significant role in amplifying the inflammatory response [100]. One of the susceptibility genes of inflammatory bowel disease (IBD) is NOD2 [100]. NO-responsive up-regulation of miR-146a expression is caused by NOD2 signaling. However, the specific mechanism of SHH signaling in IBD needs further research [100].

In summary, miR-146a plays a key role in impairing the synaptic transmission of ASD, inhibiting the migration of neurons, and contributing to the inflammatory response, but the exact mechanism needs to be studied further. Additionally, miR-146a is a promising diagnostic biomarker candidate and potential therapeutic target of ASD, when taking into consideration its abnormal expression in both the brain tissues and body fluids of ASD patients (as to be shown in Table 2, Table 3 and Table 4).

As shown in Table 1, Valleau and Sullivan found that leptin receptors (LEPRs), highly expressed in the hippocampus, may play a key role in learning and memory [111]. Since the inhibition of the JAK-STAT signal is alleviated in autistic patients [112], this suggests that the JAK-STAT signaling pathway plays a key role in the immune dysfunction of ASD. The expression of miR-153 is reduced in ASD mouse models, and miR-153 activates the JAK-STAT signaling pathway by directly increasing LEPRs [71].

PLK2 is a target of brain-specific miR-219-5p [59]. The reduced level of miR-219 will lead to overexpression of PLK2 in ASD patients. Furthermore, the overexpression of PLK2 may lead to an overall reduction in synaptic strength and neuronal excitability, which may lead to synaptic dysfunction in ASD [59].

Fragile X mental retardation gene 1 (FMR1) is a direct target of miR-221. The absence of the FMR1-encoded protein, fragile X mental retardation protein (FMRP), will cause clinical features of fragile X syndrome (FXS), including ASD behaviors [72]. FMR1 was severely up-regulated in premutated individuals who might develop the fragile X-associated tremor/ataxia syndrome (FXTAS) [72]. The overexpression of miR-221 might be a therapy for FXTAS by reducing the FMR1 expression level in brains carrying the premutation and possibly delaying its aggregation in nuclear inclusions and the appearance of disease symptoms [72].

As shown in Table 1, it is worth noting that the 3’-UTR of ARID1B mRNA can be directly bound by miR-486-3p, which reduces its protein expression through translational inhibition in SH-SY5Y cells [73]. De novo deletion of ARID1B in autism may be related to the pathogenesis of autism [74,75,76]. Furthermore, autism-related behaviors and intellectual impairment are related to synaptic dysplasia [113]. 

As shown in Table 1, the signaling pathways of EGFR and FGFR play important roles in regulating the proliferation of neural stem cells in the brain [24]. As a primate-specific miRNA, hsa_can_1002-m is down-regulated in ASD and activates EGFR and FGFR signaling pathways in the ASD cortex, including EPS8, ADAM12, CHUK, and RUNX1. Therefore, the EGFR and FGFR signaling pathways regulated by hsa_can_1002-m may play a potential role in ASD molecular pathology [24].

In addition to studies of ASD patients, some studies also examined miRNAs in ASD animal models. Genetic association studies have shown that the phenotype of ASD will be affected by some genetic mutations [52]. Therefore, FMR1, MECP2, SHANK2, and other genes are used to carry out studies in animal models to explore the pathogenesis of ASD. In a study of the ASD mouse model, it was found that miR-125b can regulate the expression of FMR1. The up-regulation of miR-125b will reduce the expression of FMR1, and FMR1 knockdown helps to change synaptic plasticity [114]. The down-regulation of miR-132 will increase the expression of MECP2, which is related to analgesic response, and miR-132 is also related to synaptic plasticity [114,115,116]. In other mouse model studies, Su et al. and Taganov et al. observed that miR-146a and miR-146b target the 3’-UTR of IRAK1. The down-regulation of miR-146a and miR-146b will increase the expression of IRAK1, thereby inducing ASD brain inflammation [117,118]. It has been reported that the down-regulation of miR-137 will lead to ASD learning and memory deficits [115]. Furthermore, the up-regulation of miR-137 can reduce the expression level of the SHANK2 gene, which will affect the postsynaptic structure [119,120]. In a valproic acid (VPA) induced rat model study, it was found that miR-30d and miR-181c were up-regulated in VPA rats and autistic amygdala, respectively, and that miR-181c regulated genes, including Akap5, ApoE, Grasp, Notch1, Ngr1, and S100b, can affect the synaptic plasticity of neurons [121,122,123]. In another VPA induced rat model study, miR-134-5p and miR-138-5p inhibited the morphogenesis of dendritic spines and vertebral columns [124,125]. Dai et al. observed that the expression of BCL2 was regulated by miR-34a in VPA rats, and it is also believed that the BCL2 signaling pathway regulated by miR-34a may play a key role in the activation of ASD [126]. These studies observed that several miRNAs, including miR-125b, miR-132, miR-137, miR-30d, and miR-181c, indeed reduce the synaptic structure and neural development in ASD animal models, thereby contributing to the pathogenesis of ASD.

In summary, a few miRNAs have been linked to ASD by targeting some relevant genes in ASD. Some miRNAs in animal models of ASD were also reported by targeting genes mainly related to synaptic functions and brain inflammation.

## 3. Dysregulated miRNAs in Brain Tissues of ASD Patients

As shown in Table 2, Abu-Elneel et al. detected 466 miRNAs in 13 ASD patients and 13 non-autistic postmortem cerebellar cortical tissues [103]. Compared with the normal controls, 28 miRNAs were found to be dysregulated, including 12 miRNAs that were up-regulated and 16 miRNAs that were down-regulated [103]. Mor et al. selected 24 postmortem brain tissue samples from an area of the frontal cortex, Brodmann area (BA) 10, including 12 ASDs and 12 controls, and identified 20 significantly dysregulated miRNAs, 16 miRNAs with up-regulated expression, and four miRNAs with down-regulated expression [68]. Ander et al. collected 18 samples from brain regions of the superior temporal sulcus (STS) + primary auditory cortex (PAC) (BA 22/41/42), including 10 ASDs and eight normal controls, and found six significantly dysregulated miRNAs, of which miR-664-3p, miR-4709-3p, and miR-4753-5p were up-regulated, and miR-1, miR-297, and miR-4742-3p were down-regulated [104]. Wu et al. collected 56 tissue samples from the cerebellar cortex (BA 9) after death, with 28 ASDs and 28 control samples. Thirty-seven dysregulated miRNAs were identified, with five down-regulated miRNAs and 32 up-regulated miRNAs [24]. Nguyen et al. collected 11 samples from temporal lobe brain tissue, with five ASDs and six controls, and found that miR-146a was up-regulated [92]. In summary, miR-146a is up-regulated in the frontal cortex (BA 10) [68] and temporal lobe [92], so it can be expected that miR-146a may play an important role in ASD. As shown in Figure 1a, miR-21-3p and miR-155-5p were up-regulated in the frontal cortex (BA 10) [68] and cerebellar cortex (BA 9) [24], indicating that miR-21-3p may have an important role in ASD, and the specific role of miR-21-3p in ASD is shown in Table 1, but the role of miR-155 in ASD is unclear [53]. In addition, as shown in Figure 1b, there are no common down-regulated miRNAs in the frontal cortex (BA 10) [68], STS + PAC (BA 22/41/42) [104], cerebellum [103], or cerebellar cortex (BA 9) [24], which might be caused by unique expression patterns of miRNAs in different brain regions. 

Stamova et al. collected 28 samples of STS and PAC, including 12 typical developing brain (TYP) control samples (six STS and six PAC), and 16 ASD samples (eight STS and eight PAC) [127]. Stamova et al. studied the correlation between the miRNAs of TYP samples and their ages and found that 18 and 27 dysregulated miRNAs in TYP_STS and TYP_PAC were associated with the ages of the individuals, respectively. Meanwhile, in the study of the correlation between miRNAs and the ages of ASD samples, they found that four and two dysregulated miRNAs in ASD_STS and ASD_PAC were associated with age, respectively. Among the four dysregulated miRNAs in ASD_STS, miR-1260b was positively correlated with age, while miR-424-3p, miR-484, and miR-3916 were negatively correlated with age [127]. The two dysregulated miRNAs in ASD_PAC, i.e., miR-93-3p and miR-3607-5p, were negatively correlated with age [127]. This research indicates that the numbers of miRNAs with dynamic expression during developmental procedures significantly decreased compared to normal controls, suggesting that the changes in miRNAs in the STS in ASD were related to social impairment, which may contribute to the molecular pathogenic mechanism of ASD [127].

In summary, different brain regions of ASD patients have different sets of dysregulated miRNAs. Only a few brain regions of ASD were profiled to identify dysregulated miRNAs. Even for the same brain region, ASD patients of different ages may have different dysregulated miRNAs.

## 4. Dysregulated miRNAs in Serum of ASD Patients

As shown in Table 3, Mundalil Vasu et al. compared the serum samples of 55 autistic patients with 55 age- and gender-matched controls, and identified 14 differentially expressed miRNAs in autistic individuals, of which six miRNAs were up-regulated and eight miRNAs were down-regulated [105]. Yu et al. collected 43 serum samples from 20 ASD patients and 23 controls, and identified two up-regulated miRNAs, namely miR-486-3p and miR-557 [73]. As shown in Table 3, we did not find miRNAs that were up-regulated in the two ASD serum sample research sets, which may be caused by using serum samples from different age groups. Mundalil Vasu et al. used samples from 6–16 year olds [105], Yu et al. used 20 ASD samples from 6.0 ± 2.8 year olds [73], and Yu et al. used 23 control samples from 5.5 ± 2.7 year olds [73]. Because serum has good stability in disease diagnosis [128], dysregulated miRNAs in serum may potentially be employed as biomarkers for the diagnosis of ASD.

To summarize, a few miRNAs in serum samples of ASD were found to have different expression levels. However, two existing studies reported different results, which might be caused by the different ages of the samples collected in these two studies.

## 5. Dysregulated miRNAs in Lymphoblastoid Cell Lines (LCLs) and Peripheral Blood of ASD Patients

As shown in Table 3, Talebizadeh et al. compared the distribution of miRNA in lymphoblastoid cell line (LCL) samples of six autistic individuals and six matched controls, and observed nine dysregulated miRNAs out of 470 expressed ones [106]. Among them, six miRNAs, i.e., miR-23a, miR-23b, miR-132, miR-146a, miR-146b, and miR-663, were up-regulated; and three miRNAs, i.e., miR-92, miR-320, and miR-363, were down-regulated [106]. Ghahramani Seno et al. extracted 42 samples from the lymphoblastoid cell line, with 20 ASDs and 22 controls, and identified 16 dysregulated miRNAs, of which 12 miRNAs were up-regulated and four miRNAs were down-regulated [58]. In an independent study using lymphoblast cell line samples, Sarachana et al. found significant differences in 43 miRNAs compared to the control group [59]. 

Huang et al. collected 40 peripheral blood samples, 20 ASDs and 20 controls, and found 44 dysregulated miRNAs, of which 20 miRNAs were down-regulated and 24 miRNAs were up-regulated [57]. As shown in Table 4, Nguyen et al. collected 14 samples, eight ASDs and six controls, from olfactory mucosal stem cells and skin fibroblasts or peripheral blood mononuclear cells, and found four dysregulated miRNAs, of which miR-146a was up-regulated and miR-221, miR-654-5p, and miR-656 expression was down-regulated [70]. Jyonouchi et al. collected 96 samples from peripheral blood mononuclear cells, with 69 ASD and 27 control samples, and identified 68 dysregulated miRNAs, with 25 up-regulated miRNAs and 43 down-regulated miRNAs [107]. Nakata et al. identified that miR-6126 was significantly down-regulated in ASD from peripheral blood analysis of ASD and control samples, and affected the severity of social defects [129]. Enrichment analysis showed that miR-6126 was associated with neural synapses and the oxytocin signaling pathway. Therefore, it is speculated that the dysregulation of miR-6126 may affect the pathogenesis of ASD [129]. Kichukova et al. collected 60 tissue samples, with 30 ASDs and 30 normal controls, from blood samples and found 40 dysregulated miRNAs, of which 11 were up-regulated and 29 were down-regulated [130]. Williams et al. collected 128 samples, with 48 ASDs and 80 controls, from blood and identified that miR-873-5p miRNA expression was up-regulated [108]. Popov et al. compared 30 autistic patients and 25 age- and sex-matched whole blood miRNA control samples and identified the down-regulation of miR-486-3p, which is a brain-specific miRNA that may play a role in the brain [131].

As shown in Figure 1c, the above research results indicate that miR-146a was up-regulated in the lymphoblastoid cell line (LCL) [106] and peripheral blood [70], which further indicates the importance of miR-146a in ASD, as shown in Table 1. In addition, as shown in Figure 1c, miR-486-3p was up-regulated in serum [73] and the LCL [58]. The specific role of miR-486-3p in ASD is shown in Table 1. As shown in Figure 1c, there are no common up-regulated miRNAs in the two studies of LCLs [58,106], which may be due to the different ages of LCL samples used in these studies. Talebizadeh et al. used child samples [106], while the ages of the samples used by Ghahramani Seno et al. are unclear [58]. As shown in Figure 1c, miR-23a and miR-23b were up-regulated in the two studies of LCLs [59,106]. As shown in Figure 1c, miR-106b and miR-186 were up-regulated in the studies of LCLs [59] and peripheral blood [107], suggesting their relevance in ASD. In addition, potential target genes of miR-23a and miR-106b are associated with neurological diseases [59]. It is shown in Figure 1c that there are no common up-regulated miRNAs in the two studies of peripheral blood of ASD patients [70,107], which may be due to the different ages of the peripheral blood samples used in these studies. Nguyen et al. used adult samples (individuals of at least 30 years of age) [70], while the median age range of samples used by Jyonouchi et al. is 10–12 years [107]. As shown in Figure 1d, miR-221 was down-regulated in the two studies of peripheral blood [70,107], and the specific role of miR-221 in ASD is shown in Table 1. The miRNA miR-320a was down-regulated in the two studies of peripheral blood [107] and serum [105], and Mundalil Vasu et al. suggest that miR-320a might be used as a noninvasive biomarker candidate for ASD [105]. In addition, miR-656 was down-regulated in the two studies of peripheral blood [70] and LCLs [58]. However, the specific role of miR-656 in ASD is still unclear.

In summary, four and three studies investigated dysregulated miRNAs in blood and LCL samples of ASD, respectively. Only a few dysregulated miRNAs were commonly identified in these studies, potentially due to the different ages of the samples in different studies.

## 6. Dysregulated miRNAs in Saliva of ASD Patients

As shown in Table 4, Toma et al. obtained 1309 samples from saliva or blood lymphocytes, including 636 ASDs and 673 controls, and identified 10 dysregulated miRNAs [109]. Hicks et al. identified 24 dysregulated miRNAs by comparing 24 ASD and 21 normal control saliva samples, of which 10 miRNAs were up-regulated and four miRNAs were down-regulated [110]. As shown in Figure 1e, we found that there are no common up-regulated miRNAs in the two studies of saliva [109,110]. Toma et al. aimed to investigate the impact of miRNA gene mutations on ASD, so 350 SNPs of 163 miRNAs were identified from saliva or blood samples of ASD [109]. Hicks et al. screened 14 dysregulated miRNAs from saliva samples and found that these miRNAs play key roles in the diagnosis of ASD and could be used as potential biomarkers for ASD [110]. Saliva also has good stability and indication in disease diagnosis [132], so the salivary dysregulated miRNAs may be used as effective biomarker candidates in the diagnosis of ASD.

In summary, two different studies found a few dysregulated miRNAs in saliva samples of ASD, respectively, but they did not report common dysregulated miRNAs.

## 7. miRNAs as Potential Diagnostic Biomarkers of ASD

Due to the stability and detectability of ASD in most tissues, miRNAs could potentially be used as biomarkers of disease. Based on the knowledge-oriented bioinformatics model, Shen et al. thought that 11 miRNAs could be used as candidate biomarkers of autism, including miR-193b-3p, miR-186-5p, miR-486-5p, miR-129-5p, miR-106b-5p, miR-181b-5p, miR-34a-5p, miR-96-5p, miR-211-5p, miR-205-5p, and miR-195-5p [25]. The serum miRNA expression profiling by qRT-PCR of 42 serum samples revealed that dysregulation of these miRNAs showed significant expression changes in children with ASD, so it is considered that serum miR-424-5p, miR-197-5p, miR-328-3p, miR-500a-5p, miR-619-5p, miR-3135a, miR-664a-3p, and miR-365a-3p might be potential biomarkers of ASD [130]. It was reported that hepatocyte growth factor (HGF) is down-regulated in the serum of ASD children, so it is considered that serum HGF concentration might be a biomarker of children with autism [133]. Compared with the control group, Vaccaro et al. found seven dysregulated miRNAs, of which miR-34c-5p, miR-92a-2-5p, miR-145-5p, and miR-199a-5p were up-regulated, while miR-27a-3p, miR-19b-1-5p, and miR-193a-5p were down-regulated in ASD patients’ blood, so it is considered that these blood dysregulated miRNAs might be used as biomarkers of ASD [134] and may regulate the pathogenesis of ASD through epigenetics. Cirnigliaro et al. found that miR-140-3p was significantly up-regulated in the serum of ASD patients compared with the control group, so it could be used as a potential serum biomarker to distinguish healthy people and ASD ysedpatients [135]. In addition, there were five serum miRNAs, miR-181b-5p, miR-320a, miR-572, miR-19b-3p, and miR-130a-3p which were very useful in ASD subject prediction, so they could potentially be used as ASD biomarkers [105]. As shown in Figure 2a, there are no common dysregulated miRNAs in one blood [134] and three serum [105,130,135] studies, which may be caused by the different age distribution of the blood and serum samples. Vaccaro et al. used child samples, with an average age of 7.5 years [134], Mundalil Vasu et al. used samples of 6–16 year olds [105], samples used by Cirnigliaro et al. were from 3–13 year olds [135], and samples used by Kichukova et al. were from 3–11 year olds, with an average age of 6.86 years [130]. The above research results indicate that these serum and blood dysregulated miRNAs may potentially be used as diagnostic biomarkers of ASD.

Salivary miRNAs can also be used as potential biomarkers of ASD. Hicks et al. identified differential miRNAs from 381 children, including 187 in ASD, 125 in typical development (TD), and 69 in non-autistic developmental delay (DD), and found 14 differentially expressed miRNAs between the ASD, TD, and DD groups [136]. Four differentially expressed miRNAs, including miR-28-3p, miR-148a-5p, miR-151a-3p, and miR-125b-2-3p, could potentially be used to identify normal individuals and ASD children, and 10 miRNAs can affect the restrictive repetitive behaviors of ASD [136]. Hicks et al. concluded that the differentially expressed salivary miRNAs in children with ASD can affect their behavior and also identify ASD patients and normal individuals [136]. Through the miRNA expression profiling of 14 autistic salivary samples by qRT-PCR, Sehovic et al. found that six salivary miRNAs were differentially expressed in ASD children and TD children, and also identified five miRNAs, including miR-7-5p, miR-23a-3p, miR-32-5p, miR-140-3p, and miR-628-5p, as potential biomarkers of ASD children [137]. As shown in Figure 2b, there are no common dysregulated miRNAs in the two studies of saliva biomarkers [136,137], which may be caused by the different age distributions of the saliva samples. Hicks et al. used samples from 2–6 year olds [136], while Sehovic et al. used samples from children with an average age of 5.8 years [137]. These findings suggest that these dysregulated salivary miRNAs may potentially be used as biomarkers of ASD to distinguish ASD and normal individuals.

Due to the potential of extracellular RNA (exRNA) as a biomarker and for the treatment of diseases, 5309 exRNA-seq data from cerebrospinal fluid (CSF), saliva, serum, plasma, and urine were collected and analyzed by exRNA Atlas [138]. For example, through exRNA Atlas analysis of the original RNA-seq data, it was revealed that urine miRNA could potentially be used as a biomarker of chronic kidney disease, gastric cancer, and myocardial infarction. In conclusion, exRNAs might be used as promising disease biomarkers, which provide practical methods for disease diagnosis.

To summarize, some miRNAs found in three serum, one blood, and two saliva sample sets could potentially be employed as biomarkers for the diagnosis of ASD. However, different studies reported different sets of miRNAs.

## 8. miRNAs as Potential Therapies for ASD

There is still no miRNA-based therapy for ASD. Presumably, the problems of miRNA delivery to brain tissues due to the blood–brain barrier, the dosage to be delivered, and side effects still need to be solved. However, some miRNA-based therapies for cancer have been designed and are being examined in clinical trials.

At present, there are two main ways of employing miRNAs for the treatment of diseases. The first way is to inhibit the overexpression of pathogenic miRNAs by delivering anti-miRNAs [139] and the second is to deliver miRNAs to compensate for the down-regulation of beneficial miRNAs [139]. In a study of mouse breast cancer models, Yoo et al. found that the combination of locked nucleic acids (LNAs) and adriamycin can inhibit miR-10b to better inhibit the growth of tumors [140]. In addition, the substances delivered through LNA nanoparticles will not affect normal tissues and have no toxicity [140]. Park et al. reported that, by delivering cholesterol-modified antimiR-221 intravenously in hepatocellular carcinoma (HCC) xenografts, miR-221 expression was down-regulated, while p27^KIP1^, p57^KIP2^, and PTEN expression of miR-221 were consequently increased, thereby contributing to mouse tumor volume shrinkage and an increase in survival time [141]. However, due to the lack of strict toxicity data of this cholesterol-modified anti-miR, its application in tumors is limited [141]. It has been reported that anti-let-7a was delivered to T87 glioblastoma by convection-enhanced delivery (CED), and the level of HMGA2 targeted by let-7a was significantly up-regulated [142]. Cortez et al. found that the survival period of animal models could be significantly prolonged when treated with miR-200c mimics in 1,2-dioleoyl-sn-glycero-3 phosphatidylcholine (DOPC) by a liposome carrier [143]. In addition, miR-200c can also target the oxidative stress response protein, which leads to an increase in ROS levels, and finally promotes the apoptosis of cancer cells [143]. It has been reported that the expression of miR-29 is down-regulated in pulmonary fibrosis. The level of endogenous miR-29 can be restored by the intravenous injection of RNA duplexes, and the function of miR-29 can be restored under the induction of bleomycin, so as to reduce the expression of collagen and reverse the pulmonary fibrosis [144]. Based on the high expression of miR-26a in normal tissues and the low expression in hepatoma, the overexpression of miR-26a delivered by adenovirus can inhibit the proliferation of mouse hepatoma model cells and induce the apoptosis of hepatoma, but it does not cause the apoptosis of healthy cells [145].

Currently, some miRNA-based therapies for cancer have entered clinical trial phases. Two companies have completed phase I and phase II clinical trials of antimiR-122. The miRNA miR-122 can maintain the level of hepatitis C virus RNA. Through the complementary combination of LNA-modified oligonucleotide (SPC3649), miR-122 can inhibit the level of hepatitis C virus RNA in chimpanzees without any side effects [146]. The antimiR-155 therapeutic agent can target cutaneous T-cell lymphoma and mycosis through LNA-modified antisense inhibitors. Phase I clinical trials are in progress. It has been reported that the combination of pH low insertion peptide (pHLIP) and antimiR-155 can help to deliver specific antimiRs to cancer cells, and this combination can prolong the survival period of mice, without obvious immunotoxicity [147]. Another miRNA-based therapeutic agent, a miR-34 mimic, targets multiple solid tumors through an lipid nanoparticles (LNPS) (Smarticles) delivery system. At present, phase I clinical trials have been completed. The miR-34 mimic delivered by liposomes can significantly inhibit tumor growth in the non-small-cell lung cancer (NSCLC) mouse model [148].

In summary, some miRNAs have played an increasingly important role in the treatment of cancer, in particular, some miRNA-based tumor therapies are being tested in clinical trials. It is expected that miRNA-based ASD therapies will appear in the future.

## 9. Conclusions

The miRNAs play critical roles in the development of the brain and maintain the homeostasis of the brain. Existing studies show that miRNAs are related to ASD, as they target several key genes of ASD. Some miRNAs also show different expression levels in brain tissues, blood samples, and even the saliva of ASD patients compared to normal controls. This raises the possibility of employing miRNAs, especially those in peripheral blood or saliva samples, as potential diagnostic biomarkers for ASD. Since different studies often reported different sets of dysregulated miRNAs, the miRNA biomarker candidates of ASD should be carefully chosen. Due to the difficulty in delivery, there is still a lack of feasible miRNA-based therapies for ASD. However, some miRNA-based therapies for cancer have entered clinical trials, and similar methods for treating ASD are expected in the near future.

## Figures and Tables

**Figure 1 ijms-21-05904-f001:**
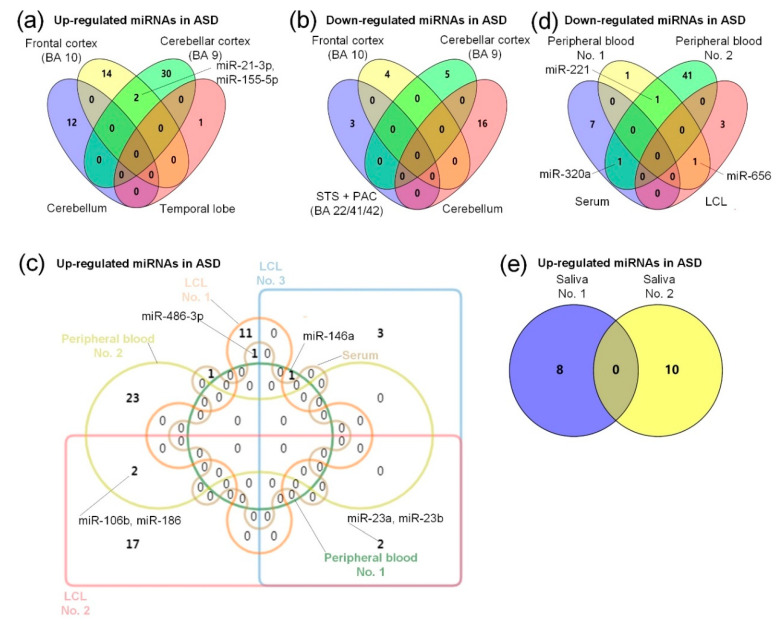
Comparisons of dysregulated miRNAs in brain and body fluid samples of ASD patients. (**a**) Summary of the up-regulated miRNA expression in four different regions of the brain. The four brain regions are the cerebellar cortex (BA 9) [24], frontal cortex (BA 10) [68], temporal lobe [92], and cerebellum [103]. (**b**) Summary of the down-regulated miRNA expression in four different brain tissues. The four brain tissues are the cerebellar cortex (BA 9) [24], frontal cortex (BA 10) [68], cerebellum [103], and STS + PAC (BA 22/41/42) [104]. (**c**) Summary of the up-regulated miRNAs in two peripheral blood, one serum [73], and three lymphoblastoid cell line (LCL) sample sets, and the two peripheral blood sample sets in the figure are divided into peripheral blood No. 1 [70], and peripheral blood No. 2 [107], and the three LCL sample sets are divided into LCL No. 1 [58], LCL No. 2 [59], and LCL No. 3 [106]. (**d**) Summary of the down-regulated miRNAs in two peripheral blood, one serum [105], and one LCL [58] sample sets, and the two peripheral blood sample sets are divided into peripheral blood No. 1 [70], and peripheral blood No. 2 [107]. (**e**) A comparison of up-regulated miRNAs in two sets of salivary samples of ASD, and the two saliva sample sets are divided into saliva No. 1 [109], and saliva No. 2 [110].

**Figure 2 ijms-21-05904-f002:**
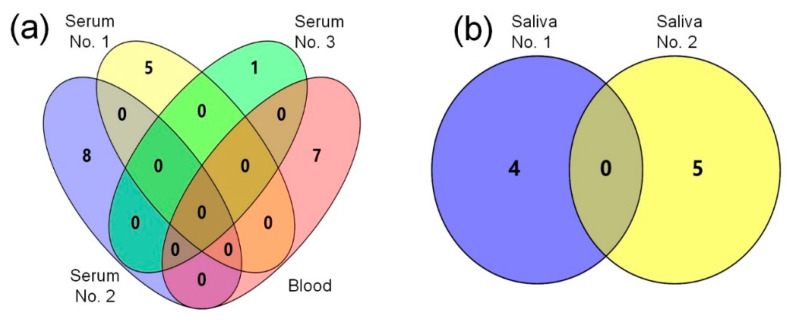
The blood, serum, and salivary miRNAs as potential biomarkers in different studies of ASD. The miRNAs identified in different studies are summarized with a Venn diagram. (**a**) A comparison of miRNAs in one blood [134] and three serum sample sets as potential biomarkers of ASD. And three serum sample sets are respectively represented by serum No. 1 [105], serum No. 2 [130], and serum No. 3 [135]. (**b**) A comparison of miRNAs as potential biomarkers in two salivary sample sets of ASD. And these two salivary miRNA sample sets are represented by saliva No. 1 [136], and saliva No. 2 [137].

**Table 1 ijms-21-05904-t001:** The miRNAs with verified functions in ASD. In the column of Change in ASD, “↑”, “↓”, and “↕” mean up-regulation, down-regulation, and inconsistent change, respectively.

miRNA	Change in ASD	Target	Function	References
miR-21-3p	↑	PAFAH1B1/LIS1,DYNC1I1	miR-21-3p inhibits the PAFAH1B1/LIS1 and DYNC1I1 genes of M16 mRNA module, which is related to neuronal migration and synapses of ASD	[24,61]
miR-21-5p	↑	OXTR	miR-21-5p inhibits OXTR translation which may aggravate ASD phenotype	[68]
miR-29b	↑	COL6A2, ID3	The inhibition of COL6A2 caused by the up-regulation of miR-29b is one of the underlying genetic mechanisms of ASD muscle disease and dyskinesia; miR-29b targets the ID3 gene. In addition, miR-29b is related to circadian rhythm signals, and studies have reported that ASD is associated with circadian rhythm disturbances	[59]
miR-103a-3p	↕	BDNF	miR-103a-3p targets the brain-derived neurotrophic factor (BDNF) gene, BDNF is directly or indirectly involved in ASD, and BDNF plays a key role in neuronal differentiation and synapses	[69]
miR-146a	↑	GRIA3, MAP1B, KCNK2	miR-146a inhibits MAP1B, GRIA3, and KCNK2, therefore impairing ASD synaptic transmission and inhibiting neuronal migration; miR-146a contributes to neuroinflammation of ASD patients	[53,70]
miR-153	↓	LEPR	The expression of miR-153 is reduced in ASD mouse models, and miR-153 activates the janus kinase-signal transducer and activator of transcription ( JAK-STAT ) signaling pathway by directly increasing leptin receptor ( LEPR), and finally attenuating the symptoms of ASD in mice	[71]
miR-219	↓	PLK2	miR-219 can directly target PLK2, and PLK2 overexpression may lead to an overall reduction in synaptic strength and neuronal excitability, which may lead to synaptic dysfunction in ASD	[59]
miR-221	↓	FMR1	miR-221 represses FMR1 at the synapse	[70,72]
miR-486-3p	↑	ARID1B	miR-486-3p can directly target ARID1B, and the mutation of ARID1B may increase the risk of ASD	[73,74,75,76]
hsa_can_1002-m	↓	EPS8, ADAM12, CHUK, RUNX1	hsa_can_1002-m activates epidermal growth factor receptor (EGFR) and fibroblast growth factor receptor (FGFR) signaling pathways in the ASD cortex by targeting EPS8, ADAM12, CHUK, and RUNX1. The EGFR and FGFR signaling pathways may play a potential role in the molecular pathology of ASD	[24]

**Table 2 ijms-21-05904-t002:** Dysregulated miRNAs in different brain regions of ASD patients.

Tissue	Samples (ASD/Control)	Up-Regulated miRNA	Down-Regulated miRNA	Reference
Cerebellar cortex (BA 9)	56 (28/28)	let-7g-3p, miR-10a-5p, miR-18b-5p, miR-20b-5p, miR-21-3p, miR-23a-3p, miR-107, miR-129-2-3p, miR-130b-5p, miR-148a-3p, miR-155-5p, miR-218-2-3p, miR-221-3p, miR-223-3p, miR-335-3p, miR-363-3p, miR-424-3p, miR-424-5p, miR-425-3p, miR-449b-5p, miR-450b-5p, miR-484, miR-629-5p, miR-651-5p, miR-708-5p, miR-766-3p, miR-874-3p, miR-887-3p, miR-940, miR-1277-3p, miR-3938, miR-2277-5p	miR-204-3p, miR-491-5p, miR-619-5p, miR-3687, miR-5096	[24]
Frontal cortex (BA 10)	24 (12/12)	miR-7-5p, miR-19a-3p, miR-19b-3p, miR-21-3p, miR-21-5p, miR-142-3p, miR-142-5p, miR-144-3p, miR-146a-5p, miR-155-5p, miR-219-5p, miR-338-5p, miR-379-5p, miR-451a, miR-494, miR-3168	miR-34a-5p, miR-92b-3p, miR-211-5p, miR-3960	[68]
Temporal lobe	11 (5/6)	miR-146a		[92]
Postmortem cerebellar tissue	26 (13/13)	miR-106a, miR-106b, miR-140, miR-146b, miR-181d, miR-193b, miR-320a, miR-381, miR-432, miR-539, miR-550, miR-652	miR-7, miR-15a, miR-15b, miR-21, miR-23a, miR-27a, miR-93, miR-95, miR-128, miR-129, miR-132, miR-148b, miR-212, miR-431, miR-484, miR-598	[103]
STS + PAC (BA 22/41/42)	18 (10/8)	miR-664-3p, miR-4709-3p, miR-4753-5p	miR-1, miR-297, miR-4742-3p	[104]

**Table 3 ijms-21-05904-t003:** Dysregulated miRNAs in serum and lymphoblastoid cell lines (LCLs) of ASD patients.

Tissue	Samples (ASD/Control)	Up-Regulated miRNA	Down-Regulated miRNA	Reference
Serum	43 (20/23)	miR-486-3p, miR-557		[73]
Serum	110 (55/55)	miR-19b-3p, miR-27a-3p, miR-101-3p, miR-106-5p, miR-130a-3p, miR-195b-5p	miR-151a-3p, miR-181b-5p, miR-320a, miR-328, miR-433, miR-489, miR-572, miR-663a	[105]
Lymphoblastoid cell lines (LCLs)	42 (20/22)	miR-10a, miR-30a, miR-181a, miR-181b, miR-181c, miR-199b-5p, miR-338-3p, miR-486-3p, miR-486-5p, miR-500, miR-502-3p, miR-548	miR-199a-5p, miR-455-3p, miR-577, miR-656	[58]
Lymphoblastoid cell lines (LCLs)	14 (5/9)	miR-23a, miR-23b, miR-25, miR-29b, miR-30c, miR-93, miR-103, miR-106b, miR-107, miR-185, miR-186, miR-191, miR-194, miR-195, miR-205, miR-342, miR-346, miR-376a-AS, miR-451, miR-519c, miR-524	miR-16-2, miR-106b, miR-132, miR-133b, miR-136, miR-139, miR-148b, miR-153-1, miR-182-AS, miR-189, miR-190, miR-199b, miR-211, miR-219, miR-326, miR-367, miR-455, miR-495, miR-518a, miR-520b	[59]
Lymphoblastoid cell lines (LCLs)	12 (6/6)	miR-23a, miR-23b, miR-132, miR-146a, miR-146b, miR-663	miR-92, miR-320, miR-363	[106]
Peripheral blood	40 (20/20)	miR-34b-3p, miR-34c-3p, miR-483-5p, miR-494, miR-564, miR-574-5p, miR-575, miR-642a-3p, miR-921, miR-1246, miR-1249, miR-1273c, miR-4270, miR-4299, miR-4436a, miR-4443, miR-4516, miR-4669, miR-4721, miR-4728-5p, miR-4788, miR-5739, miR-6086, miR-6125	let-7a-5p, let-7d-5p, let-7f-5p, miR-15a-5p, miR-15b-5p, miR-16-5p, miR-19b-3p, miR-20a-5p, miR-92a-3p, miR-103a-3p, miR-195-5p, miR-451a, miR-574-3p, miR-940, miR-1228-3p, miR-3613-3p, miR-3935, miR-4436b-5p, miR-4665-5p, miR-4700-3p	[57]

**Table 4 ijms-21-05904-t004:** Dysregulated miRNAs in blood and saliva of ASD patients.

Tissue	Samples (ASD/Control)	Up-Regulated miRNA	Down-Regulated miRNA	Reference
Peripheral blood	14 (8/6)	miR-146a	miR-221, miR-654-5p, miR-656	[70]
Peripheral blood	96 (69/27)	let-7a-1, let-7a-2, let-7a-3, let-7f-1, let-7f-2, let-7g, let-7i, miR-17, miR-26a-2, miR-30b, miR-30c-1, miR-30c-2, miR-98, miR-106b, miR-130a, miR-148a, miR-148b, miR-150, miR-186, miR-301a, miR-374b, miR-494, miR-1248, miR-3607, miR-3609	let-7b, miR-15a, miR-15b, miR-16-1, miR-16-2, miR-18a, miR-19a, miR-19b-1, miR-19b-2, miR-20a, miR-21, miR-27a, miR-27b, miR-29a, miR-29b-1, miR-29b-2, miR-29c, miR-30e, miR-93, miR-101-1, miR-101-2, miR-103a-1, miR-103a-2, miR-107, miR-126, miR-142, miR-145, miR-146a, miR-151a, miR-181a-1, miR-181a-2, miR-199b, miR-221, miR-222, miR-320a, miR-376c, miR-409, miR-423, miR-484, miR-625, miR-4433b, miR-5701-1, miR-5701-2	[107]
Blood	128 (48/80)	miR-873-5p		[108]
Saliva or blood lymphocytes	1309 (636/673)	miR-17, miR-18a, miR-19a, miR-20a, miR-19b-1, miR-92a-1, miR-133b, miR-206		[109]
Saliva	45 (24/21)	miR-7-5p, miR-28-5p, miR-127-3p, miR-140-3p, miR-191-5p, miR-218-5p, miR-335-3p, miR-628-5p, miR-2467-5p, miR-3529-3p	miR-23a-3p, miR-27a-3p, miR-30e-5p, miR-32-5p	[110]

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
