# Peer review of "Recent Progress on Relevant microRNAs in Autism Spectrum Disorders"

_ijms, 2020, doi:10.3390/ijms21165904_

Round 1

Reviewer 1 Report

In this review, the authors summarized the data regarding changes in microRNAs in patients with ASD. Changes were found in the brain, serum, saliva, lymphoblastoid cells, and urine. The authors claim that microRNAs can possibly be used as biomarkers for ASD, yet they show that different investigators described changes in different microRNAs, hence there are not unanimously agreed microRNAs that can be biomarkers. This issue must be clarified as it remind us of the situation regarding the changes in hundreds of genes observed in patients with ASD, yet none of these genes can serve as a marker (except in few defined syndromes with ASD like behavior). The authors must also give a short summary after each section, to show the reader what we can learn from the data shown. They also must explain why they added some animal data, and what we can learn from that about the human situation. The conclusions should also be changed if the microRNAs cannot yet serve as markers and also to include some mention of the fact that different investigators observed different changes. 

The English need improvement and the referral to summaries on different topics in the introduction should be shorter. 

Author Response

Comments 1.1 In this review, the authors summarized the data regarding changes in microRNAs in patients with ASD. Changes were found in the brain, serum, saliva, lymphoblastoid cells, and urine. The authors claim that microRNAs can possibly be used as biomarkers for ASD, yet they show that different investigators described changes in different microRNAs, hence there are not unanimously agreed microRNAs that can be biomarkers. This issue must be clarified as it remind us of the situation regarding the changes in hundreds of genes observed in patients with ASD, yet none of these genes can serve as a marker (except in few defined syndromes with ASD like behavior).

Responses 1.1 This review summarized the dysregulated miRNAs in body fluids and several brain regions. At present, it is true that no dysregulated miRNAs can be used as biomarkers for ASD in body fluids and brain regions. We have revised the related statements in the manuscript. For examples, we changed the title of Section 7, “miRNAs as diagnostic biomarkers of ASD” to “miRNAs as potential diagnostic biomarkers of ASD”; “Due to the stability and detectability of ASD in most tissues, miRNAs are used as biomarkers of disease” to “Due to the stability and detectability of ASD in most tissues, miRNAs could potentially be used as biomarkers of disease”.

Comments 1.2 The authors must also give a short summary after each section, to show the reader what we can learn from the data shown.

Responses 1.2

Thanks for your comments. We have added summary paragraphs for all sections except Introduction and Conclusion. For your convenience, we also put these summary paragraphs below.

Section 2 miRNAs with characterized functions in ASD

In summary, a few miRNAs have been linked to ASD by targeting some relevant genes in ASD. Some miRNAs in animal models of ASD were also reported by targeting genes mainly related to synaptic functions and brain inflammation.

Section 3 Dysregulated miRNAs in brain tissues of ASD patients

In summary, different brain regions of ASD patients have different sets of dysregulated miRNAs. Only a few brain regions of ASD were profiles to identify dysregulated miRNAs. Even for the same brain region, the ASD patients of different ages may have different dysregulated miRNAs.

Section 4 Dysregulated miRNAs in serum of ASD patients

To summarize, a few miRNAs in serum samples of ASD were found to have different expression levels. However, two existing studies reported different results, which might be caused by the different ages of the samples collected in these two studies.

Section 5 Dysregulated miRNAs in lymphoblastoid cell lines (LCL) and peripheral blood of ASD patients

In summary, four and three studies investigated dysregulated miRNAs in blood and LCL samples of ASD, respectively. Only a few dysregulated miRNAs were commonly identified in these studies, potentially due to the different ages of samples in different studies.

Section 6 Dysregulated miRNAs in saliva of ASD patients

In summary, two different studies found a few dysregulated miRNAs in Serum samples of ASD, respectively, but they did not report common dysregulated miRNAs.

Section 7 miRNAs as diagnostic biomarkers of ASD

To summarize, some miRNAs found from 3 Serum, 1 blood, and 2 Saliva sample sets could potentially be employed as biomarkers for diagnosis of ASD. However, different studies reported different sets of miRNAs.

Section 8 miRNAs as potential therapies for ASD

In summary, some miRNAs have played an increasingly important role in the treatment of cancer, especially some miRNA-based tumor therapies are being tested in clinical trials. It is expected that miRNA-based ASD therapies would appear in the foreseeable future.

Comments 1.3 They also must explain why they added some animal data, and what we can learn from that about the human situation.

Responses 1.3 Since it is infeasible to directly employ patients for molecular experiments of ASD, animal models were used to examine the molecular mechanism related to ASD. These experiments and results from animal models will be of great help to clarify the important pathogenic mechanism of ASD and to test the effects of different treatments.

Comments 1.4 The conclusions should also be changed if the microRNAs cannot yet serve as markers and also to include some mention of the fact that different investigators observed different changes.

Responses 1.4 We have revised the conclusion to mention that some of dysregulated miRNAs in the blood, serum and saliva samples of ASD may only be used as potential biomarkers for ASD. And we added “Since different studies often reported different sets of dysregulated miRNAs, the miRNAs as biomarker candidates of ASD should be carefully chosen.”

Comments 1.5 The English need improvement and the referral to summaries on different topics in the introduction should be shorter.

Responses 1.5

Thank you for your carefulness. We carefully read and revised the whole manuscript. A lot of grammatical and typing errors were corrected in the revised manuscript. Some of the main corrected were also included below for your convenience.

We also revised the Introduction to remove some duplicate statements and combined the Sections 1-3 in the last submission.

  1. Page 1, line 5, “Kunming University of Science and Technology, Kunming 650000, China” -> “Kunming University of Science and Technology, Kunming, Yunnan 650500, China”;
  2. Page 1, line 19, “the sum of AIDS, cancer and diabetes” -> “the sum of AIDS, cancer, and diabetes”;
  3. Page 2, line 7, “and it was estimated that 1 in 59 children is ASD patient in 2018” -> “and it was estimated that 1 in 59 children is affected by ASD in 2018”;
  4. Page 2, line 20, “miRNAs were selected as biomarkers to improve the diagnosis, prognosis and treatment of ASD” -> “miRNAs were selected as biomarkers to improve the diagnosis, prognosis, and treatment of ASD”;
  5. Page 2, line 21, “In addition” -> “Furthermore”;
  6. Page 2, line 23, “the life quality of ASD patients have been significantly improved though the effort that has been made by the whole society” -> “the life quality of ASD patients has been significantly improved through the effort that has been made by the whole society”;
  7. Page 2, line 25, “there are still some ASD patients who have difficult in living independently” -> “there are still some ASD patients who have difficulty in independent living”;
  8. Page 3, line 1, “4. miRNAs with characterized functions in ASD” -> “2. miRNAs with characterized functions in ASD”;
  9. Page 3, line 2, “The MAPK signaling pathway in which the candidate target genes of both miR-let-7a and miR-let-7d involves in, directly or indirectly associated with the physiopathology of ASD” -> “The MAPK signaling pathway in which the candidate target genes of both let-7a and let-7d involves, is directly or indirectly associated with the physiopathology of ASD”;
  10. Page 3, line 7, “It has been found that miR-let-7a is highly expressed in in early human embryonic tissues” -> “It has been found that let-7a is highly expressed in early human embryonic tissues”;
  11. Page 3, line 8, “which suggests the key role miR-let-7a plays in early embryonic development [61]. So, though the embryonic development process miR-let-7a may be indirectly associated with ASD” -> “which suggests the key role let-7a plays in early embryonic development [61]. Therefore, through the embryonic development process, let-7a may be indirectly associated with ASD”;
  12. Page 4, line 1, “Table 1. miRNAs with verified functions in ASD” -> “Table 1. miRNAs with verified functions in ASD. In the column of Change in ASD, “↑”, “↓”, and “↕” means up-regulation, down-regulation and inconsistent change, respectively”;
  13. Page 5, line 1, “↑, miR-103a-3p represses BDNF which improves neuron and synapse function of ASD” -> “, miR-103a-3p targets the BDNF gene, and BDNF is directly or indirectly involved in ASD, and BDNF plays a key role in neuronal differentiation and synapses”;
  14. Page 5, line 4, “miR-146a inhibits MAP1B, GRIA3 and KCNK2 target genes, thereby impairing ASD synaptic transmission” -> “miR-146a inhibits MAP1B, GRIA3, and KCNK2, therefore impairing ASD synaptic transmission”;
  15. Page 6, line 4, “miR-486-3p can directly target the 3ʹ-UTR of ARID1B” -> “miR-486-3p can directly target ARID1B”;
  16. Page 6, line 9, “hsa_can_1002-m activates EGFR and FGFR signaling pathways in the ASD cortex, including EPS8, ADAM12, CHUK, and RUNX1” -> “hsa_can_1002-m activates EGFR and FGFR signaling pathways in the ASD cortex, by targeting EPS8, ADAM12, CHUK, and RUNX1”;
  17. Page 6, line 10, “and the EGFR and FGFR signaling pathways involved in these genes may play a potential role in the molecular pathology of ASD” -> “And the EGFR and FGFR signaling pathways may play a potential role in the molecular pathology of ASD”;
  18. Page 7, line 2, “miR-21-3p inhibits multiple genes in M16 gene module on 15q11-13” -> “miR-21-3p inhibits multiple genes in the M16 gene module on 15q11-13”;
  19. Page 7, line 5, “These three genes play keys role in the migration of neurons [62]. In conclusion, M16 module gene is related to neuronal migration and synapse of ASD” -> “These three genes play key roles in the migration of neurons [62]. In conclusion, the M16 module gene is related to neuronal migration and synapse of ASD”;
  20. Page 7, line 6, “In addition” -> “Besides”;
  21. Page 7, line 7, “the overexpression of miR-21-3p leads to a significant decrease in PCDH19” -> “the overexpression of miR-21-3p leads to a significant decrease in the expression of PCDH19”;
  22. Page 7, line 13, “miR-21-5p, is also relevant in ASD. In ASD brain” -> “miR-21-5p is also relevant in ASD. In the ASD brain”;
  23. Page 7, line 15, “miR-21-5p may aggravate the phenotype of ASD by reducing the expression of OXTR in ASD brain” -> “miR-21-5p may aggravate the phenotype of ASD by reducing the expression of OXTR in the ASD brain”;
  24. Page 7, line 18, “mutation of COL6A2 results in reduction of COL6A2 transcripts” -> “Mutation of COL6A2 results in reduction of COL6A2 transcripts”;
  25. Page 7, line 24, “The precursor of miR‐29b leads to the down‐regulation of ID3 transcript” -> “The overexpression of miR-29b leads to the down-regulation of the ID3 transcript [59]”;
  26. Page 7, line 37, “and is highly expressed in brain regions such as cortex, hippocampus and amygdala” -> “and is highly expressed in brain regions such as cortex, hippocampus, and amygdala”;
  27. Page 7, line 39, “miR-146a inhibits the expression of neuronal specific genes, Nlgn1 and Syt1” -> “miR-146a inhibits the expression of neuronal-specific genes, Nlgn1, and Syt1”;
  28. Page 7, line 40, “In brain of mouse model” -> “In the brain of the mouse model”;
  29. Page 7, line 41, “resulting in shrunken dendrite” -> “resulting in shrinking dendrite”;
  30. Page 8, line 4, “up-regulation of miR-146a in ASD neurons” -> “the up-regulation of miR-146a in ASD neurons”;
  31. Page 8, line 7, “up-regulation of miR-146a in ASD” -> “the up-regulation of miR-146a in ASD”;
  32. Page 8, line 11, “Up‐regulation of miR‐146a occurs in the early childhood brain of ASD” -> “The up‐regulation of miR‐146a occurs in the early childhood brain of ASD”;
  33. Page 8, line 13, “overexpression of miR-146a results in neurite outgrowth, branching enhancement, imbalance between neural progenitor cell renewal and neuronal differentiation” -> “overexpression of miR-146a results in neurite outgrowth, branching enhancement, the imbalance between neural progenitor cell renewal, and neuronal differentiation”;
  34. Page 8, line 15, “Autistic patients have signs of nerve inflammation, immune abnormality and changes in the inflammatory response” -> “Autistic patients have signs of nerve inflammation, immune abnormality, and changes in the inflammatory response”;
  35. Page 8, line 19, “miR-146a is one of the most common dysregulated miRNA in the development brain disorders (DBD) patients including ASD” -> “miR-146a is one of the most common dysregulated miRNAs in developmental brain disorders (DBD) patients including ASD”;
  36. Page 8, line 27, “The activation of M1 macrophages is inhibited by miR-146a” -> “Activation of M1 macrophages is inhibited by miR-146a”;
  37. Page 8, line 33, “NUMB plays key role in the asymmetric division of neural progenitor cells” -> “NUMB plays a key role in the asymmetric division of neural progenitor cells”;
  38. Page 8, line 34, “miR-146a targets the NUMB gene, thereby activating Sonic hedgehog (SHH) signaling expression [95]” -> “miR-146a targets the NUMB gene, thereby activating the Sonic hedgehog (SHH) signaling expression [95]”;
  39. Page 8, line 40, “However, the specific mechanism of SHH signaling in IBD needs further research to clarify” -> “However, the specific mechanism of SHH signaling in IBD needs further research”;
  40. Page 8, line 41, “miR-146a play keys role in impairing the synaptic transmission of ASD” -> “miR-146a plays a key role in impairing the synaptic transmission of ASD”;
  41. Page 8, line 44, “And miR-146a is a promising diagnostic biomarker and therapeutic target of ASD” -> “And miR-146a is a promising diagnostic biomarker candidate and potential therapeutic target of ASD”;
  42. Page 9, line 1, “as to be shown in Table 2, 3 and 4” -> “as to be shown in Tables 2, 3, and 4”;
  43. Page 9, line 4, “Valleau and Sullivan found that leptin receptors (LEPR), highly expressed in the hippocampus, plays a key role in learning and memory” -> “Valleau and Sullivan found that leptin receptors (LEPR), highly expressed in the hippocampus may play a key role in learning and memory”;
  44. Page 9, line 5, “Since the inhibition of JAK-STAT signal is alleviated in autistic patients [99]” -> “Since the inhibition of the JAK-STAT signal is alleviated in autistic patients [99]”;
  45. Page 9, line 6, “the JAK-STAT signaling pathway plays a key role in immune dysfunction of ASD” -> “suggesting that the JAK-STAT signaling pathway plays a key role in the immune dysfunction of ASD”;
  46. Page 9, line 7, “and miR-153 activates JAK-STAT signaling pathway by directly increasing LEPR” -> “and miR-153 activates the JAK-STAT signaling pathway by directly increasing LEPR”;
  47. Page 9, line 9, “In addition, the reduced level of miR-219 will lead to overexpression of PLK2 in ASD patients,” -> “The reduced level of miR-219 will lead to overexpression of PLK2 in ASD patients.”;
  48. Page 9, line 10, “and PLK2 overexpression may lead to an overall reduction in synaptic strength and neuronal excitability” -> “Furthermore, overexpression of PLK2 may lead to an overall reduction in synaptic strength and neuronal excitability”;
  49. Page 9, line 15, “including ASD behavior [101]. FMR1 is severely upregulated in premutated individuals” -> “including ASD behaviors [101]. FMR1 was severely up-regulated in premutated individuals”;
  50. Page 9, line 33, “Therefore, FMR1, MECP2, SHANK2 and other genes are used to carry out studies in animal models to explore the pathogenesis of ASD” -> “Therefore, FMR1, MECP2, SHANK2, and other genes are used to carry out studies in animal models to explore the pathogenesis of ASD”;
  51. Page 9, line 35, “In the study of ASD mouse model, miR-125b can regulate the expression of FMR1 gene” -> “In the study of the ASD mouse model, miR-125b can regulate the expression of FMR1”;
  52. Page 9, line 40, “Su et al. and Taganov et al. observed that miR-146a and miR-146b target the 3ʹ-UTR sequence of IRAK1 gene” -> “Su et al. and Taganov et al. observed that miR-146a and miR-146b target the 3ʹ-UTR of IRAK1”;
  53. Page 9, line 44, “Furthermore, the up-regulation of miR-137 can reduce the expression level of SHANK2 gene, which will affect the structure of postsynaptic” -> “Furthermore, the up-regulation of miR-137 can reduce the expression level of the SHANK2 gene, which will affect the postsynaptic structure”;
  54. Page 10, line 2, “and miR-181c regulate genes, including Akap5, ApoE, Grasp, Notch1, Ngr1, S100b” -> “and miR-181c regulates genes, including Akap5, ApoE, Grasp, Notch1, Ngr1, S100b”;
  55. Page 10, line 8, “including miR-125b, miR-132, miR-137, miR-30d and miR-181c” -> “including miR-125b, miR-132, miR-137, miR-30d, and miR-181c”;
  56. Page 10, line 9, “indeed reduce the synaptic structure and neural development of ASD animal models” -> “indeed reduce the synaptic structure and neural development in ASD animal models”;
  57. Page 10, line 11, “5. Dysregulated miRNAs in brain tissues of ASD patients” -> “3. Dysregulated miRNAs in brain tissues of ASD patients”;
  58. Page 12, line 7, “Ander et al. collected 18 samples from brain regions of STS and PAC, BA 22/41/42, including 10 ASDs and 8 normal controls” -> “Ander et al. collected 18 samples from brain regions of superior temporal sulcus (STS) + primary auditory cortex (PAC) (BA 22/41/42), including 10 ASDs and 8 normal controls”;
  59. Page 12, line 10, “Wu et al. collected 56 tissue samples from the cerebellar cortex, BA 9 after death with 28 ASDs and 28 control samples” -> “Wu et al. collected 56 tissue samples from the cerebellar cortex (BA 9) after death with 28 ASDs and 28 control samples”;
  60. Page 12, line 14, “In summary, miR-146a is up-regulated in frontal cortex, BA 10 [69] and temporal lobe [87]” -> “In summary, miR-146a is up-regulated in the frontal cortex (BA 10) [69] and temporal lobe [87]”;
  61. Page 12, line 16, “miR-21-3p and miR-155-5p were up-regulated in frontal cortex, BA 10 [69] and cerebellar cortex, BA 9 [31]” -> “miR-21-3p and miR-155-5p were up-regulated in the frontal cortex (BA 10) [69] and cerebellar cortex (BA 9) [31]”;
  62. Page 12, line 19, “there are no common down-regulated miRNAs in frontal cortex, BA 10 [69], STS and PAC, BA 22/41/42 [121], cerebellar [120], cerebellar cortex, BA 9 [31]” -> “there are no common down-regulated miRNAs in the frontal cortex (BA 10) [69], STS + PAC (BA 22/41/42) [121], cerebellar [120], cerebellar cortex (BA 9) [31]”;
  63. Page 13, line 2, “The four brain regions are Frontal cortex, BA 10, Temporal lobe, Cerebellar, and Cerebellar cortex, BA 9” -> “The four brain regions are the Frontal cortex (BA 10), Temporal lobe, Cerebellar, and Cerebellar cortex (BA 9)”;
  64. Page 13, line 4, “The four brain tissues are Frontal cortex, BA 10, STS and PAC, BA 22/41/42, Cerebellar and Cerebellar cortex, BA 9” -> “The four brain tissues are the Frontal cortex (BA 10), STS + PAC (BA 22/41/42), Cerebellar, and Cerebellar cortex (BA 9)”;
  65. Page 13, line 6, “Summary of the up-regulated miRNAs in Peripheral blood, serum and Lymphoblastoid cell lines (LCL)” -> “Summary of the up-regulated miRNAs in two Peripheral blood, one Serum, and three Lymphoblastoid Cell Lines (LCL) sample sets”;
  66. Page 13, line 7, “Summary of the down-regulated miRNAs in Peripheral blood, serum and LCL” -> “Summary of the down-regulated miRNAs in two Peripheral blood, one Serum, and one LCL sample sets”;
  67. Page 13, line 8, “A comparison of up-regulated miRNAs in saliva samples of two studies” -> “A comparison of up-regulated miRNAs in two sets of Saliva samples of ASD”;
  68. Page 13, line 9, “Stamova et al. collected 28 samples of superior temporal sulcus (STS) and primary audit cortex (PAC)” -> “Stamova et al. collected 28 samples of STS and PAC”;
  69. Page 13, line 15, “they found that 4 and 2 dysregulated miRNAs in ASD_STS and ASD_PAC are associated with the ages” -> “they found that 4 and 2 dysregulated miRNAs in ASD_STS and ASD_PAC were associated with the ages”;
  70. Page 13, line 16, “Among the 4 dysregualted miRNAs in ASD_STS, miR-1260b are positively correlated with the ages” -> “Among the 4 dysregulated miRNAs in ASD_STS, miR-1260b was positively correlated with the ages”;
  71. Page 13, line 17, “while miR-424-3p, miR-484, miR-3916 are negatively correlated with ages” -> “while miR-424-3p, miR-484, and miR-3916 were negatively correlated with ages [121]”;
  72. Page 13, line 18, “The two dysregulated miRNAs in ASD_PAC, i.e., miR-93-3p and miR-3607-5p, are negatively correlated with age” -> “The two dysregulated miRNAs in ASD_PAC, i.e., miR-93-3p and miR-3607-5p were negatively correlated with age”;
  73. Page 13, line 21, “the changes of miRNAs in the superior temporal sulcus (STS) in ASD are related to their social impairment” -> “the changes of miRNAs in the STS in ASD were related to their social impairment”;
  74. Page 13, line 24, “6. Dysregulated miRNAs in serum of ASD patients” -> “4. Dysregulated miRNAs in serum of ASD patients”;
  75. Page 15, line 6, “dysregulated miRNAs in serum may be employed as biomarkers for the diagnosis of ASD” -> “dysregulated miRNAs in serum may potentially be employed as biomarkers for the diagnosis of ASD”;
  76. Page 15, line 8, “7. Dysregulated miRNAs in lymphoblastoid cell lines (LCL) and peripheral blood of ASD patients” -> “5. Dysregulated miRNAs in lymphoblastoid cell lines (LCL) and peripheral blood of ASD patients”;
  77. Page 17, line 1, “Huang et al. collected 40 peripheral blood samples, 20 ASDs and 20 controls and found 44 dysregulated miRNAs” -> “Huang et al. collected 40 peripheral blood samples, 20 ASDs, and 20 controls and found 44 dysregulated miRNAs”;
  78. Page 17, line 11, “Enrichment analysis showed that miR-6126 is associated with neural synapse and oxytocin signaling pathway,” -> “Enrichment analysis showed that miR-6126 was associated with neural synapse and oxytocin signaling pathway.”;
  79. Page 17, line 19, “which is a brain specific miRNA that may play a role in the brain” -> “which is a brain-specific miRNA that may play a role in the brain”;
  80. Page 17, line 21, “which further indicate the importance of miR-146a in ASD” -> “which further indicates the importance of miR-146a in ASD”;
  81. Page 17, line 24, “there is no common up-regulated miRNA in the two studies of LCL” -> “there are no commonly up-regulated miRNAs in the two studies of LCL”;
  82. Page 17, line 30, “In addition, the potential target genes of miR-23a and miR-106b are associated with neurological diseases” -> “In addition, potential target genes of miR-23a and miR-106b are associated with neurological diseases”;
  83. Page 17, line 31, “It is shown in Figure 1c that there is no common up-regulated miRNA in the two studies of peripheral blood of ASD patients” -> “It is shown in Figure 1c that there are no commonly up-regulated miRNAs in the two studies of peripheral blood of ASD patients”;
  84. Page 17, line 39, “Vasu et al. suggest that miR-320a might be used as a noninvasive biomarker for ASD” -> “Vasu et al. suggest that miR-320a might be used as a noninvasive biomarker candidate for ASD”;
  85. Page 17, line 41, “8. Dysregulated miRNAs in saliva of ASD patients” -> “6. Dysregulated miRNAs in saliva of ASD patients”;
  86. Page 18, line 7, “Hicks et al. screened 14 dysregulated miRNAs from saliva samples and found that these miRNAs play a key role in the diagnosis of ASD and could be used as biomarkers for ASD” -> “Hicks et al. screened 14 dysregulated miRNAs from saliva samples and found that these miRNAs play key roles in the diagnosis of ASD and could be used as potential biomarkers for ASD”;
  87. Page 18, line 10, “so the salivary dysregulated miRNAs may be used as effective biomarkers in the diagnosis of ASD” -> “so the salivary dysregulated miRNAs may be used as effective biomarker candidates in the diagnosis of ASD”;
  88. Page 18, line 11, “ miRNAs as diagnostic biomarkers of ASD” -> “7. miRNAs as potential diagnostic biomarkers of ASD”;
  89. Page 18, line 12, “miRNAs are used as biomarkers of disease” -> “miRNAs could potentially be used as biomarkers of disease”;
  90. Page 18, line 25, “so it is considered that these blood dysregulated miRNAs might be biomarkers of ASD” -> “so it is considered that these blood dysregulated miRNAs might be used as biomarkers of ASD”;
  91. Page 18, line 30, “there are 5 serum miRNAs, miR-181b-5p, miR-320a, miR-572, miR-19b-3p and miR-130a-3p” -> “there are 5 serum miRNAs, miR-181b-5p, miR-320a, miR-572, miR-19b-3p, and miR-130a-3p”;
  92. Page 18, line 31, “so they can be used as ASD biomarkers” -> “so they could potentially be used as ASD biomarkers”;
  93. Page 18, line 34, “Vasu et al. used age distribution of samples was 6-16 years old” -> “Vasu et al. used samples of 6-16 years old”;
  94. Page 18, line 35, “Cirnigliaro et al. used age distribution of samples was 3-13 years old” -> “samples used by Cirnigliaro et al. were 3-13 years old”;
  95. Page 18, line 36, “Kichukova et al. used age distribution of samples was 3-11 years old ,” -> “samples used by Kichukova et al. were 3-11 years old,”;
  96. Page 18, line 38, “The above research results indicate that these serum and blood dysregulated miRNAs may be used as biomarkers of ASD and can be used for its diagnosis” -> “The above research results indicate that these serum and blood dysregulated miRNAs may potentially be used as diagnostic biomarkers of ASD”.
  97. Page 19, line 1, “The blood, serum and saliva miRNA biomarkers in different studies of ASD” -> “The Blood, Serum, and Salivary miRNA biomarkers in different studies of ASD”;
  98. Page 19, line 2, “The miRNAs identified in different studiers were summarized with the Venn diagram” -> “The miRNAs identified in different studies were summarized with the Venn diagram”;
  99. Page 19, line 3, “A comparison of miRNAs in blood and serum as biomarkers of ASD” -> “A comparison of miRNAs in Blood and Serum as biomarkers of ASD”;
  100. Page 19, line 4, “A comparison of miRNAs in saliva as biomarkers of ASD” -> “A comparison of Salivary miRNAs as biomarkers of ASD”;
  101. Page 19, line 5, “Salivary miRNAs can also be used as biomarkers of ASD” -> “Salivary miRNAs can also be used as potential biomarkers of ASD”;
  102. Page 19, line 8, “and found 14 differentially expressed miRNAs between the ASD, TD and DD groups” -> “and found 14 differentially expressed miRNAs between the ASD, TD and DD groups [136]”;
  103. Page 19, line 9, “including miR-28-3p, miR-148a-5p, miR-151a-3p and miR-125b-2-3p can be used to identify normal individuals and ASD children” -> “including miR-28-3p, miR-148a-5p, miR-151a-3p, and miR-125b-2-3p could potentially be used to identify normal individuals and ASD children”;
  104. Page 19, line 10, “and 10 miRNAs can affect the restrictive repetitive behaviors of ASD” -> “and 10 miRNAs can affect the restrictive repetitive behaviors of ASD [136]”;
  105. Page 19, line 13, “Though 14 autistic salivary miRNAs expression profiling by qRT-PCR” -> “Through miRNA expression profiling of 14 autistic salivary samples by qRT-PCR”;
  106. Page 19, line 16, “miR-628-5p as potential biomarkers of ASD children, that can be used to distinguish ASD children and normal individuals” -> “miR-628-5p as potential biomarkers of ASD children [137]”;
  107. Page 19, line 21, “These findings suggest that these dysregulated salivary miRNAs may be used as biomarkers of ASD to diagnose ASD and normal individuals, and can also identify dysregulated miRNAs that affect ASDʹs restrictive repetitive behaviors” -> “These findings suggest that these dysregulated salivary miRNAs may potentially be used as biomarkers of ASD to diagnose ASD and normal individuals”;
  108. Page 19, line 25, “5,309 exRNA-seq data from cerebrospinal fluid (CSF), serum, plasma and urine were collected and analyzed by exRNA Atlas” -> “5,309 exRNA-seq data from cerebrospinal fluid (CSF), serum, plasma, and urine were collected and analyzed by exRNA Atlas”;
  109. Page 19, line 28, “it reveals that urine miRNA can be used as a biomarker of chronic kidney disease; through exRNA atlas analysis of the original RNA-seq data, it reveals that small RNA can be used as a new biomarker of gastric cancer, and small RNA can also be used as a biomarker of myocardial infarction” -> “it reveals that urine miRNA could potentially be used as a biomarker of chronic kidney disease, gastric cancer, and myocardial infarction”;
  110. Page 20, line 1, “which provide practical methods for disease diagnosis” -> “which provides practical methods for disease diagnosis”;
  111. Page 20, line 2, “10. miRNAs as potential therapies for ASD” -> “8. miRNAs as potential therapies for ASD”;
  112. Page 20, line 4, “the problems of miRNA delivery to brain tissues due to blood-brain barrier” -> “the problems of miRNA delivery to brain tissues due to the blood-brain barrier”;
  113. Page 20, line 8, “The first way is to inhibit the over expression of pathogenic miRNAs by delivering anti-miRNAs” -> “The first way is to inhibit the overexpression of pathogenic miRNAs by delivering anti-miRNAs”;
  114. Page 20, line 11, “Yoo et al. found that the combination of LNA” -> “Yoo et al. found that the combination of locked nucleic acids (LNAs)”;
  115. Page 20, line 12, “and adriamycin can inhibit miR-10b to better inhibit the growth of tumor” -> “and adriamycin can inhibit miR-10b to better inhibit the growth of tumors”.
  116. Page 20, line 20, “and the level of HMGA2 targeted by let-7a was significantly upregulated” -> “and the level of HMGA2 targeted by let-7a was significantly up-regulated”;
  117. Page 20, line 34, “Through the complementary combination of locking nucleic acid (LNA) modified oligonucleotide (SPC3649)” -> “Through the complementary combination of LNA modified oligonucleotide (SPC3649)”;
  118. Page 20, line 38, “It has been reported that the combination of pHLIp and antimiR-155 can help to deliver antimir specific to cancer cells” -> “It has been reported that the combination of pHLIP and antimiR-155 can help to deliver specific antimiR to cancer cells”;
  119. Page 21, line 3, “ Conclusion” -> “9. Conclusion”;
  120. Page 21, line 4, “miRNAs play critical roles in the development of brain and maintain the homeostasis of brain” -> “miRNAs play critical roles in the development of the brain and maintain the homeostasis of the brain”;
  121. Page 21, line 7, “Some miRNAs also show different expression levels in brain tissues, blood samples and even saliva of ASD patients compared to normal controls” -> “Some miRNAs also show different expression levels in brain tissues, blood samples, and even saliva of ASD patients compared to normal controls”;
  122. Page 21, line 9, “These raise the possibility of employing miRNAs, especially in peripheral blood or saliva samples, for diagnosis of ASD” -> “These raise the possibility of employing miRNAs, especially in peripheral blood or saliva samples, as potential diagnostic biomarkers for ASD”.

Reviewer 2 Report

This review depicts the possible roles of miRNAs in Autism as biomarkers and possible drug targets. The manuscript is well structured and is written a clear scientific language. In addition, the authors compare the distinct miRNA profiles and indicate that the research on miRNAs still remain important in order to achieve more precise sets of autism specific miRNAs for future adaptation in clinics. Therefore, I suggest to accept the manuscript after better English spell check. 

Author Response

Comments 2.1 This review depicts the possible roles of miRNAs in Autism as biomarkers and possible drug targets. The manuscript is well structured and is written a clear scientific language. In addition, the authors compare the distinct miRNA profiles and indicate that the research on miRNAs still remain important in order to achieve more precise sets of autism specific miRNAs for future adaptation in clinics. Therefore, I suggest to accept the manuscript after better English spell check.

Responses 2.1

Thank you for your comments. We have carefully read and revised the manuscript to correct grammatical errors and typos as best as we could. Some of the major revisions are listed below for your convenience.

  1. Page 1, line 5, “Kunming University of Science and Technology, Kunming 650000, China” -> “Kunming University of Science and Technology, Kunming, Yunnan 650500, China”;
  2. Page 1, line 19, “the sum of AIDS, cancer and diabetes” -> “the sum of AIDS, cancer, and diabetes”;
  3. Page 2, line 7, “and it was estimated that 1 in 59 children is ASD patient in 2018” -> “and it was estimated that 1 in 59 children is affected by ASD in 2018”;
  4. Page 2, line 20, “miRNAs were selected as biomarkers to improve the diagnosis, prognosis and treatment of ASD” -> “miRNAs were selected as biomarkers to improve the diagnosis, prognosis, and treatment of ASD”;
  5. Page 2, line 21, “In addition” -> “Furthermore”;
  6. Page 2, line 23, “the life quality of ASD patients have been significantly improved though the effort that has been made by the whole society” -> “the life quality of ASD patients has been significantly improved through the effort that has been made by the whole society”;
  7. Page 2, line 25, “there are still some ASD patients who have difficult in living independently” -> “there are still some ASD patients who have difficulty in independent living”;
  8. Page 3, line 1, “4. miRNAs with characterized functions in ASD” -> “2. miRNAs with characterized functions in ASD”;
  9. Page 3, line 2, “The MAPK signaling pathway in which the candidate target genes of both miR-let-7a and miR-let-7d involves in, directly or indirectly associated with the physiopathology of ASD” -> “The MAPK signaling pathway in which the candidate target genes of both let-7a and let-7d involves, is directly or indirectly associated with the physiopathology of ASD”;
  10. Page 3, line 7, “It has been found that miR-let-7a is highly expressed in in early human embryonic tissues” -> “It has been found that let-7a is highly expressed in early human embryonic tissues”;
  11. Page 3, line 8, “which suggests the key role miR-let-7a plays in early embryonic development [61]. So, though the embryonic development process miR-let-7a may be indirectly associated with ASD” -> “which suggests the key role let-7a plays in early embryonic development [61]. Therefore, through the embryonic development process, let-7a may be indirectly associated with ASD”;
  12. Page 4, line 1, “Table 1. miRNAs with verified functions in ASD” -> “Table 1. miRNAs with verified functions in ASD. In the column of Change in ASD, “↑”, “↓”, and “↕” means up-regulation, down-regulation and inconsistent change, respectively”;
  13. Page 5, line 1, “↑, miR-103a-3p represses BDNF which improves neuron and synapse function of ASD” -> “, miR-103a-3p targets the BDNF gene, and BDNF is directly or indirectly involved in ASD, and BDNF plays a key role in neuronal differentiation and synapses”;
  14. Page 5, line 4, “miR-146a inhibits MAP1B, GRIA3 and KCNK2 target genes, thereby impairing ASD synaptic transmission” -> “miR-146a inhibits MAP1B, GRIA3, and KCNK2, therefore impairing ASD synaptic transmission”;
  15. Page 6, line 4, “miR-486-3p can directly target the 3ʹ-UTR of ARID1B” -> “miR-486-3p can directly target ARID1B”;
  16. Page 6, line 9, “hsa_can_1002-m activates EGFR and FGFR signaling pathways in the ASD cortex, including EPS8, ADAM12, CHUK, and RUNX1” -> “hsa_can_1002-m activates EGFR and FGFR signaling pathways in the ASD cortex, by targeting EPS8, ADAM12, CHUK, and RUNX1”;
  17. Page 6, line 10, “and the EGFR and FGFR signaling pathways involved in these genes may play a potential role in the molecular pathology of ASD” -> “And the EGFR and FGFR signaling pathways may play a potential role in the molecular pathology of ASD”;
  18. Page 7, line 2, “miR-21-3p inhibits multiple genes in M16 gene module on 15q11-13” -> “miR-21-3p inhibits multiple genes in the M16 gene module on 15q11-13”;
  19. Page 7, line 5, “These three genes play keys role in the migration of neurons [62]. In conclusion, M16 module gene is related to neuronal migration and synapse of ASD” -> “These three genes play key roles in the migration of neurons [62]. In conclusion, the M16 module gene is related to neuronal migration and synapse of ASD”;
  20. Page 7, line 6, “In addition” -> “Besides”;
  21. Page 7, line 7, “the overexpression of miR-21-3p leads to a significant decrease in PCDH19” -> “the overexpression of miR-21-3p leads to a significant decrease in the expression of PCDH19”;
  22. Page 7, line 13, “miR-21-5p, is also relevant in ASD. In ASD brain” -> “miR-21-5p is also relevant in ASD. In the ASD brain”;
  23. Page 7, line 15, “miR-21-5p may aggravate the phenotype of ASD by reducing the expression of OXTR in ASD brain” -> “miR-21-5p may aggravate the phenotype of ASD by reducing the expression of OXTR in the ASD brain”;
  24. Page 7, line 18, “mutation of COL6A2 results in reduction of COL6A2 transcripts” -> “Mutation of COL6A2 results in reduction of COL6A2 transcripts”;
  25. Page 7, line 24, “The precursor of miR‐29b leads to the down‐regulation of ID3 transcript” -> “The overexpression of miR-29b leads to the down-regulation of the ID3 transcript [59]”;
  26. Page 7, line 37, “and is highly expressed in brain regions such as cortex, hippocampus and amygdala” -> “and is highly expressed in brain regions such as cortex, hippocampus, and amygdala”;
  27. Page 7, line 39, “miR-146a inhibits the expression of neuronal specific genes, Nlgn1 and Syt1” -> “miR-146a inhibits the expression of neuronal-specific genes, Nlgn1, and Syt1”;
  28. Page 7, line 40, “In brain of mouse model” -> “In the brain of the mouse model”;
  29. Page 7, line 41, “resulting in shrunken dendrite” -> “resulting in shrinking dendrite”;
  30. Page 8, line 4, “up-regulation of miR-146a in ASD neurons” -> “the up-regulation of miR-146a in ASD neurons”;
  31. Page 8, line 7, “up-regulation of miR-146a in ASD” -> “the up-regulation of miR-146a in ASD”;
  32. Page 8, line 11, “Up‐regulation of miR‐146a occurs in the early childhood brain of ASD” -> “The up‐regulation of miR‐146a occurs in the early childhood brain of ASD”;
  33. Page 8, line 13, “overexpression of miR-146a results in neurite outgrowth, branching enhancement, imbalance between neural progenitor cell renewal and neuronal differentiation” -> “overexpression of miR-146a results in neurite outgrowth, branching enhancement, the imbalance between neural progenitor cell renewal, and neuronal differentiation”;
  34. Page 8, line 15, “Autistic patients have signs of nerve inflammation, immune abnormality and changes in the inflammatory response” -> “Autistic patients have signs of nerve inflammation, immune abnormality, and changes in the inflammatory response”;
  35. Page 8, line 19, “miR-146a is one of the most common dysregulated miRNA in the development brain disorders (DBD) patients including ASD” -> “miR-146a is one of the most common dysregulated miRNAs in developmental brain disorders (DBD) patients including ASD”;
  36. Page 8, line 27, “The activation of M1 macrophages is inhibited by miR-146a” -> “Activation of M1 macrophages is inhibited by miR-146a”;
  37. Page 8, line 33, “NUMB plays key role in the asymmetric division of neural progenitor cells” -> “NUMB plays a key role in the asymmetric division of neural progenitor cells”;
  38. Page 8, line 34, “miR-146a targets the NUMB gene, thereby activating Sonic hedgehog (SHH) signaling expression [95]” -> “miR-146a targets the NUMB gene, thereby activating the Sonic hedgehog (SHH) signaling expression [95]”;
  39. Page 8, line 40, “However, the specific mechanism of SHH signaling in IBD needs further research to clarify” -> “However, the specific mechanism of SHH signaling in IBD needs further research”;
  40. Page 8, line 41, “miR-146a play keys role in impairing the synaptic transmission of ASD” -> “miR-146a plays a key role in impairing the synaptic transmission of ASD”;
  41. Page 8, line 44, “And miR-146a is a promising diagnostic biomarker and therapeutic target of ASD” -> “And miR-146a is a promising diagnostic biomarker candidate and potential therapeutic target of ASD”;
  42. Page 9, line 1, “as to be shown in Table 2, 3 and 4” -> “as to be shown in Tables 2, 3, and 4”;
  43. Page 9, line 4, “Valleau and Sullivan found that leptin receptors (LEPR), highly expressed in the hippocampus, plays a key role in learning and memory” -> “Valleau and Sullivan found that leptin receptors (LEPR), highly expressed in the hippocampus may play a key role in learning and memory”;
  44. Page 9, line 5, “Since the inhibition of JAK-STAT signal is alleviated in autistic patients [99]” -> “Since the inhibition of the JAK-STAT signal is alleviated in autistic patients [99]”;
  45. Page 9, line 6, “the JAK-STAT signaling pathway plays a key role in immune dysfunction of ASD” -> “suggesting that the JAK-STAT signaling pathway plays a key role in the immune dysfunction of ASD”;
  46. Page 9, line 7, “and miR-153 activates JAK-STAT signaling pathway by directly increasing LEPR” -> “and miR-153 activates the JAK-STAT signaling pathway by directly increasing LEPR”;
  47. Page 9, line 9, “In addition, the reduced level of miR-219 will lead to overexpression of PLK2 in ASD patients,” -> “The reduced level of miR-219 will lead to overexpression of PLK2 in ASD patients.”;
  48. Page 9, line 10, “and PLK2 overexpression may lead to an overall reduction in synaptic strength and neuronal excitability” -> “Furthermore, overexpression of PLK2 may lead to an overall reduction in synaptic strength and neuronal excitability”;
  49. Page 9, line 15, “including ASD behavior [101]. FMR1 is severely upregulated in premutated individuals” -> “including ASD behaviors [101]. FMR1 was severely up-regulated in premutated individuals”;
  50. Page 9, line 33, “Therefore, FMR1, MECP2, SHANK2 and other genes are used to carry out studies in animal models to explore the pathogenesis of ASD” -> “Therefore, FMR1, MECP2, SHANK2, and other genes are used to carry out studies in animal models to explore the pathogenesis of ASD”;
  51. Page 9, line 35, “In the study of ASD mouse model, miR-125b can regulate the expression of FMR1 gene” -> “In the study of the ASD mouse model, miR-125b can regulate the expression of FMR1”;
  52. Page 9, line 40, “Su et al. and Taganov et al. observed that miR-146a and miR-146b target the 3ʹ-UTR sequence of IRAK1 gene” -> “Su et al. and Taganov et al. observed that miR-146a and miR-146b target the 3ʹ-UTR of IRAK1”;
  53. Page 9, line 44, “Furthermore, the up-regulation of miR-137 can reduce the expression level of SHANK2 gene, which will affect the structure of postsynaptic” -> “Furthermore, the up-regulation of miR-137 can reduce the expression level of the SHANK2 gene, which will affect the postsynaptic structure”;
  54. Page 10, line 2, “and miR-181c regulate genes, including Akap5, ApoE, Grasp, Notch1, Ngr1, S100b” -> “and miR-181c regulates genes, including Akap5, ApoE, Grasp, Notch1, Ngr1, S100b”;
  55. Page 10, line 8, “including miR-125b, miR-132, miR-137, miR-30d and miR-181c” -> “including miR-125b, miR-132, miR-137, miR-30d, and miR-181c”;
  56. Page 10, line 9, “indeed reduce the synaptic structure and neural development of ASD animal models” -> “indeed reduce the synaptic structure and neural development in ASD animal models”;
  57. Page 10, line 11, “5. Dysregulated miRNAs in brain tissues of ASD patients” -> “3. Dysregulated miRNAs in brain tissues of ASD patients”;
  58. Page 12, line 7, “Ander et al. collected 18 samples from brain regions of STS and PAC, BA 22/41/42, including 10 ASDs and 8 normal controls” -> “Ander et al. collected 18 samples from brain regions of superior temporal sulcus (STS) + primary auditory cortex (PAC) (BA 22/41/42), including 10 ASDs and 8 normal controls”;
  59. Page 12, line 10, “Wu et al. collected 56 tissue samples from the cerebellar cortex, BA 9 after death with 28 ASDs and 28 control samples” -> “Wu et al. collected 56 tissue samples from the cerebellar cortex (BA 9) after death with 28 ASDs and 28 control samples”;
  60. Page 12, line 14, “In summary, miR-146a is up-regulated in frontal cortex, BA 10 [69] and temporal lobe [87]” -> “In summary, miR-146a is up-regulated in the frontal cortex (BA 10) [69] and temporal lobe [87]”;
  61. Page 12, line 16, “miR-21-3p and miR-155-5p were up-regulated in frontal cortex, BA 10 [69] and cerebellar cortex, BA 9 [31]” -> “miR-21-3p and miR-155-5p were up-regulated in the frontal cortex (BA 10) [69] and cerebellar cortex (BA 9) [31]”;
  62. Page 12, line 19, “there are no common down-regulated miRNAs in frontal cortex, BA 10 [69], STS and PAC, BA 22/41/42 [121], cerebellar [120], cerebellar cortex, BA 9 [31]” -> “there are no common down-regulated miRNAs in the frontal cortex (BA 10) [69], STS + PAC (BA 22/41/42) [121], cerebellar [120], cerebellar cortex (BA 9) [31]”;
  63. Page 13, line 2, “The four brain regions are Frontal cortex, BA 10, Temporal lobe, Cerebellar, and Cerebellar cortex, BA 9” -> “The four brain regions are the Frontal cortex (BA 10), Temporal lobe, Cerebellar, and Cerebellar cortex (BA 9)”;
  64. Page 13, line 4, “The four brain tissues are Frontal cortex, BA 10, STS and PAC, BA 22/41/42, Cerebellar and Cerebellar cortex, BA 9” -> “The four brain tissues are the Frontal cortex (BA 10), STS + PAC (BA 22/41/42), Cerebellar, and Cerebellar cortex (BA 9)”;
  65. Page 13, line 6, “Summary of the up-regulated miRNAs in Peripheral blood, serum and Lymphoblastoid cell lines (LCL)” -> “Summary of the up-regulated miRNAs in two Peripheral blood, one Serum, and three Lymphoblastoid Cell Lines (LCL) sample sets”;
  66. Page 13, line 7, “Summary of the down-regulated miRNAs in Peripheral blood, serum and LCL” -> “Summary of the down-regulated miRNAs in two Peripheral blood, one Serum, and one LCL sample sets”;
  67. Page 13, line 8, “A comparison of up-regulated miRNAs in saliva samples of two studies” -> “A comparison of up-regulated miRNAs in two sets of Saliva samples of ASD”;
  68. Page 13, line 9, “Stamova et al. collected 28 samples of superior temporal sulcus (STS) and primary audit cortex (PAC)” -> “Stamova et al. collected 28 samples of STS and PAC”;
  69. Page 13, line 15, “they found that 4 and 2 dysregulated miRNAs in ASD_STS and ASD_PAC are associated with the ages” -> “they found that 4 and 2 dysregulated miRNAs in ASD_STS and ASD_PAC were associated with the ages”;
  70. Page 13, line 16, “Among the 4 dysregualted miRNAs in ASD_STS, miR-1260b are positively correlated with the ages” -> “Among the 4 dysregulated miRNAs in ASD_STS, miR-1260b was positively correlated with the ages”;
  71. Page 13, line 17, “while miR-424-3p, miR-484, miR-3916 are negatively correlated with ages” -> “while miR-424-3p, miR-484, and miR-3916 were negatively correlated with ages [121]”;
  72. Page 13, line 18, “The two dysregulated miRNAs in ASD_PAC, i.e., miR-93-3p and miR-3607-5p, are negatively correlated with age” -> “The two dysregulated miRNAs in ASD_PAC, i.e., miR-93-3p and miR-3607-5p were negatively correlated with age”;
  73. Page 13, line 21, “the changes of miRNAs in the superior temporal sulcus (STS) in ASD are related to their social impairment” -> “the changes of miRNAs in the STS in ASD were related to their social impairment”;
  74. Page 13, line 24, “6. Dysregulated miRNAs in serum of ASD patients” -> “4. Dysregulated miRNAs in serum of ASD patients”;
  75. Page 15, line 6, “dysregulated miRNAs in serum may be employed as biomarkers for the diagnosis of ASD” -> “dysregulated miRNAs in serum may potentially be employed as biomarkers for the diagnosis of ASD”;
  76. Page 15, line 8, “7. Dysregulated miRNAs in lymphoblastoid cell lines (LCL) and peripheral blood of ASD patients” -> “5. Dysregulated miRNAs in lymphoblastoid cell lines (LCL) and peripheral blood of ASD patients”;
  77. Page 17, line 1, “Huang et al. collected 40 peripheral blood samples, 20 ASDs and 20 controls and found 44 dysregulated miRNAs” -> “Huang et al. collected 40 peripheral blood samples, 20 ASDs, and 20 controls and found 44 dysregulated miRNAs”;
  78. Page 17, line 11, “Enrichment analysis showed that miR-6126 is associated with neural synapse and oxytocin signaling pathway,” -> “Enrichment analysis showed that miR-6126 was associated with neural synapse and oxytocin signaling pathway.”;
  79. Page 17, line 19, “which is a brain specific miRNA that may play a role in the brain” -> “which is a brain-specific miRNA that may play a role in the brain”;
  80. Page 17, line 21, “which further indicate the importance of miR-146a in ASD” -> “which further indicates the importance of miR-146a in ASD”;
  81. Page 17, line 24, “there is no common up-regulated miRNA in the two studies of LCL” -> “there are no commonly up-regulated miRNAs in the two studies of LCL”;
  82. Page 17, line 30, “In addition, the potential target genes of miR-23a and miR-106b are associated with neurological diseases” -> “In addition, potential target genes of miR-23a and miR-106b are associated with neurological diseases”;
  83. Page 17, line 31, “It is shown in Figure 1c that there is no common up-regulated miRNA in the two studies of peripheral blood of ASD patients” -> “It is shown in Figure 1c that there are no commonly up-regulated miRNAs in the two studies of peripheral blood of ASD patients”;
  84. Page 17, line 39, “Vasu et al. suggest that miR-320a might be used as a noninvasive biomarker for ASD” -> “Vasu et al. suggest that miR-320a might be used as a noninvasive biomarker candidate for ASD”;
  85. Page 17, line 41, “8. Dysregulated miRNAs in saliva of ASD patients” -> “6. Dysregulated miRNAs in saliva of ASD patients”;
  86. Page 18, line 7, “Hicks et al. screened 14 dysregulated miRNAs from saliva samples and found that these miRNAs play a key role in the diagnosis of ASD and could be used as biomarkers for ASD” -> “Hicks et al. screened 14 dysregulated miRNAs from saliva samples and found that these miRNAs play key roles in the diagnosis of ASD and could be used as potential biomarkers for ASD”;
  87. Page 18, line 10, “so the salivary dysregulated miRNAs may be used as effective biomarkers in the diagnosis of ASD” -> “so the salivary dysregulated miRNAs may be used as effective biomarker candidates in the diagnosis of ASD”;
  88. Page 18, line 11, “9. miRNAs as diagnostic biomarkers of ASD” -> “7. miRNAs as potential diagnostic biomarkers of ASD”;
  89. Page 18, line 12, “miRNAs are used as biomarkers of disease” -> “miRNAs could potentially be used as biomarkers of disease”;
  90. Page 18, line 25, “so it is considered that these blood dysregulated miRNAs might be biomarkers of ASD” -> “so it is considered that these blood dysregulated miRNAs might be used as biomarkers of ASD”;
  91. Page 18, line 30, “there are 5 serum miRNAs, miR-181b-5p, miR-320a, miR-572, miR-19b-3p and miR-130a-3p” -> “there are 5 serum miRNAs, miR-181b-5p, miR-320a, miR-572, miR-19b-3p, and miR-130a-3p”;
  92. Page 18, line 31, “so they can be used as ASD biomarkers” -> “so they could potentially be used as ASD biomarkers”;
  93. Page 18, line 34, “Vasu et al. used age distribution of samples was 6-16 years old” -> “Vasu et al. used samples of 6-16 years old”;
  94. Page 18, line 35, “Cirnigliaro et al. used age distribution of samples was 3-13 years old” -> “samples used by Cirnigliaro et al. were 3-13 years old”;
  95. Page 18, line 36, “Kichukova et al. used age distribution of samples was 3-11 years old ,” -> “samples used by Kichukova et al. were 3-11 years old,”;
  96. Page 18, line 38, “The above research results indicate that these serum and blood dysregulated miRNAs may be used as biomarkers of ASD and can be used for its diagnosis” -> “The above research results indicate that these serum and blood dysregulated miRNAs may potentially be used as diagnostic biomarkers of ASD”.
  97. Page 19, line 1, “The blood, serum and saliva miRNA biomarkers in different studies of ASD” -> “The Blood, Serum, and Salivary miRNA biomarkers in different studies of ASD”;
  98. Page 19, line 2, “The miRNAs identified in different studiers were summarized with the Venn diagram” -> “The miRNAs identified in different studies were summarized with the Venn diagram”;
  99. Page 19, line 3, “A comparison of miRNAs in blood and serum as biomarkers of ASD” -> “A comparison of miRNAs in Blood and Serum as biomarkers of ASD”;
  100. Page 19, line 4, “A comparison of miRNAs in saliva as biomarkers of ASD” -> “A comparison of Salivary miRNAs as biomarkers of ASD”;
  101. Page 19, line 5, “Salivary miRNAs can also be used as biomarkers of ASD” -> “Salivary miRNAs can also be used as potential biomarkers of ASD”;
  102. Page 19, line 8, “and found 14 differentially expressed miRNAs between the ASD, TD and DD groups” -> “and found 14 differentially expressed miRNAs between the ASD, TD and DD groups [136]”;
  103. Page 19, line 9, “including miR-28-3p, miR-148a-5p, miR-151a-3p and miR-125b-2-3p can be used to identify normal individuals and ASD children” -> “including miR-28-3p, miR-148a-5p, miR-151a-3p, and miR-125b-2-3p could potentially be used to identify normal individuals and ASD children”;
  104. Page 19, line 10, “and 10 miRNAs can affect the restrictive repetitive behaviors of ASD” -> “and 10 miRNAs can affect the restrictive repetitive behaviors of ASD [136]”;
  105. Page 19, line 13, “Though 14 autistic salivary miRNAs expression profiling by qRT-PCR” -> “Through miRNA expression profiling of 14 autistic salivary samples by qRT-PCR”;
  106. Page 19, line 16, “miR-628-5p as potential biomarkers of ASD children, that can be used to distinguish ASD children and normal individuals” -> “miR-628-5p as potential biomarkers of ASD children [137]”;
  107. Page 19, line 21, “These findings suggest that these dysregulated salivary miRNAs may be used as biomarkers of ASD to diagnose ASD and normal individuals, and can also identify dysregulated miRNAs that affect ASDʹs restrictive repetitive behaviors” -> “These findings suggest that these dysregulated salivary miRNAs may potentially be used as biomarkers of ASD to diagnose ASD and normal individuals”;
  108. Page 19, line 25, “5,309 exRNA-seq data from cerebrospinal fluid (CSF), serum, plasma and urine were collected and analyzed by exRNA Atlas” -> “5,309 exRNA-seq data from cerebrospinal fluid (CSF), serum, plasma, and urine were collected and analyzed by exRNA Atlas”;
  109. Page 19, line 28, “it reveals that urine miRNA can be used as a biomarker of chronic kidney disease; through exRNA atlas analysis of the original RNA-seq data, it reveals that small RNA can be used as a new biomarker of gastric cancer, and small RNA can also be used as a biomarker of myocardial infarction” -> “it reveals that urine miRNA could potentially be used as a biomarker of chronic kidney disease, gastric cancer, and myocardial infarction”;
  110. Page 20, line 1, “which provide practical methods for disease diagnosis” -> “which provides practical methods for disease diagnosis”;
  111. Page 20, line 2, “10. miRNAs as potential therapies for ASD” -> “8. miRNAs as potential therapies for ASD”;
  112. Page 20, line 4, “the problems of miRNA delivery to brain tissues due to blood-brain barrier” -> “the problems of miRNA delivery to brain tissues due to the blood-brain barrier”;
  113. Page 20, line 8, “The first way is to inhibit the over expression of pathogenic miRNAs by delivering anti-miRNAs” -> “The first way is to inhibit the overexpression of pathogenic miRNAs by delivering anti-miRNAs”;
  114. Page 20, line 11, “Yoo et al. found that the combination of LNA” -> “Yoo et al. found that the combination of locked nucleic acids (LNAs)”;
  115. Page 20, line 12, “and adriamycin can inhibit miR-10b to better inhibit the growth of tumor” -> “and adriamycin can inhibit miR-10b to better inhibit the growth of tumors”.
  116. Page 20, line 20, “and the level of HMGA2 targeted by let-7a was significantly upregulated” -> “and the level of HMGA2 targeted by let-7a was significantly up-regulated”;
  117. Page 20, line 34, “Through the complementary combination of locking nucleic acid (LNA) modified oligonucleotide (SPC3649)” -> “Through the complementary combination of LNA modified oligonucleotide (SPC3649)”;
  118. Page 20, line 38, “It has been reported that the combination of pHLIp and antimiR-155 can help to deliver antimir specific to cancer cells” -> “It has been reported that the combination of pHLIP and antimiR-155 can help to deliver specific antimiR to cancer cells”;
  119. Page 21, line 3, “ Conclusion” -> “9. Conclusion”;
  120. Page 21, line 4, “miRNAs play critical roles in the development of brain and maintain the homeostasis of brain” -> “miRNAs play critical roles in the development of the brain and maintain the homeostasis of the brain”;
  121. Page 21, line 7, “Some miRNAs also show different expression levels in brain tissues, blood samples and even saliva of ASD patients compared to normal controls” -> “Some miRNAs also show different expression levels in brain tissues, blood samples, and even saliva of ASD patients compared to normal controls”;
  122. Page 21, line 9, “These raise the possibility of employing miRNAs, especially in peripheral blood or saliva samples, for diagnosis of ASD” -> “These raise the possibility of employing miRNAs, especially in peripheral blood or saliva samples, as potential diagnostic biomarkers for ASD”.

Round 2

Reviewer 1 Report

Accept in current form